# Two distinct binding modes provide the RNA-binding protein RbFox with extraordinary sequence specificity

Xuan Ye [1,2,8], Wen Yang[3,4,8], Soon Yi [1,2], Yanan Zhao[5], Gabriele Varani[3] ✉, Eckhard Jankowsky [1,2,6,7] ✉ & Fan Yang [5] ✉

Specificity of RNA-binding proteins for target sequences varies considerably. Yet, it is not understood how certain few proteins achieve markedly higher sequence specificity than most others. Here we show that the RNA Recognition Motif of RbFox accomplishes extraordinary sequence specificity by employing functionally and structurally distinct binding modes. Affinity measurements of RbFox for all binding site variants reveal the existence of two distinct binding modes. The first exclusively accommodates cognate and closely related RNAs with high affinity. The second mode accommodates all other RNAs with reduced affinity by imposing large thermodynamic penalties on non-cognate sequences. NMR studies indicate marked structural differences between the two binding modes, including large conformational rearrangements distant from the RNA-binding site. Distinct binding modes by a single RNA-binding module explain extraordinary sequence selectivity and reveal an unknown layer of functional diversity, cross talk and regulation in RNA-protein interactions.

In eukaryotic cells, the expression of tens of thousands of RNAs is regulated by thousands of diverse RNA-binding proteins (RBPs) that interact with RNA in a carefully orchestrated manner[1,2]. A pivotal aspect in this choreography of regulation is the specificity of RBPs for their target sites[1–7]; that is, the degree to which an RBP discriminates between cognate and non-cognate RNAs[1,2,4]. Specificity of RBPs varies considerably; some RBPs are highly selective for certain RNA sites, others bind degenerate sites, and yet other RBPs interact with RNAs broadly, with little sequence discrimination[8–14]. Defining the molecular mechanisms through which RBPs acquire their specificity is critical for understanding the rules of RNA biology and for devising therapeutic strategies against diseases associated with RBP- and RNA-related processes[1,2,15].

Among RBPs with the greatest specificity are the three closely related human RbFox proteins (RbFox 1-3), which function in pre-mRNA splicing, miRNA processing and other RNA metabolic steps[16]. The three RbFox proteins contain divergent N- and C-terminal regions that are likely unstructured but all share a highly conserved RNA-binding domain that belongs to the RNA Recognition Motif (RRM) superfamily[16,17]. The RbFox RRM binds RNAs with a consensus 5′-GCAUG motif with low nanomolar affinity in vitro[18–21]. Even minor changes in the consensus motif decrease the RNA affinity by orders of magnitude[19,21], indicating an inherent specificity of RbFox proteins for their consensus sequence that is markedly higher than for most other RBPs[8,10,11]. In the cell, however, biological effects by RbFox proteins are also

[1]Center for RNA Science and Therapeutics, School of Medicine, Case Western Reserve University, Cleveland, OH, USA. [2]Department of Biochemistry, School of Medicine, Case Western Reserve University, Cleveland, OH, USA. [3]Department of Chemistry, University of Washington, Seattle, WA, USA. [4]Greater Bay Biomedical InnoCenter, Shenzhen Bay Laboratory, Shenzhen 518055, China. [5]School of Life Science and Technology, Harbin Institute of Technology, Harbin 150080, China. [6]Case Comprehensive Cancer Center, School of Medicine, Case Western Reserve University, Cleveland, OH, USA. [7]Present address: Moderna Therapeutics, 200 Technology Square, Cambridge, MA, USA. [8]These authors contributed equally: Xuan Ye, Wen Yang. ✉e-mail: varani@chem.washington.edu; exj13@case.edu; fanyang115@hit.edu.cn

exerted by binding to non-consensus sites characterized by lower affinity, when expression levels of RbFox increase[22].

How RbFox proteins accomplish higher specificity than most other RBPs is not understood. Even the existing NMR structures, which provide detailed views of the RNA-binding interface, fail to provide a cogent reason for this extraordinary specificity[19,20]. Therefore, we set out to systematically examine the basis for the high specificity of the RbFox proteins through a combination of high throughput biochemical techniques followed by NMR structure determination. We first determined the affinity of the RRM of RbFox for all possible 16,384 7-mer RNA sequence variants. The resulting affinity distribution reveals two distinct binding modes, one associated with binding to the consensus 5′-GCAUG and a second mode for binding to all other sequences. Mutations in RbFox only have a small effect on the binding of the consensus sequence, but markedly increase affinity for all other variants, thereby diminishing specificity. The results indicate that the binding mode for the non-consensus sequences imposes a large thermodynamic penalty on these sequence variants, thereby enabling exquisite discrimination between consensus and non-consensus RNAs. Comparison of the NMR structures with a non-consensus and the consensus sequence reveals that the two binding modes are associated with substantial structural differences within the RRM. Remarkably, the structural rearrangement extends to protein regions distant from the RNA-binding interface, suggesting that the distinct binding modes can transmit RNA sequence information to potential protein binding partners of RbFox.

Thus, our data show that RbFox employs a previously undescribed mechanism to accomplish its extraordinary specificity - structurally distinct binding modes that enable the imposition of large thermodynamic penalties on non-consensus sequences and transmit RNA sequence information to distant regions of the RRM. The results reveal another layer of biological complexity in RNA-protein interactions and post-transcriptional regulation of gene expression.

## Results

### The affinity distribution of RbFox is bimodal

To examine how RbFox achieves high specificity, we first measured the affinity of the RNA-binding domain (RRM) of RbFox for two RNAs that differ by a single nucleotide (Fig. 1a). The apparent equilibrium binding constant ($K_{1/2}$) for the RbFox cognate sequence (5′-GCAUG) was reduced by orders of magnitude by this single nucleotide substitution (Fig. 1a, b), consistent with previous data[19–21]. This result highlights the remarkable ability of RbFox to discriminate effectively between RNA substrates that differ by only small sequence variations.

Next, we comprehensively characterized the specificity landscape of the RbFox RRM by employing High Throughput Sequencing Equilibrium Binding (HiTS-Eq)[12,13,23,24]. This approach allows the simultaneous determination of apparent affinities for large numbers of sequence variants within an RNA pool[12,13,23]. We constructed a pool of 16,384 unique RNAs encoding all possible 7-mer sequences in a segment with 7 randomized nucleotides (Fig. 1c). This segment was flanked on each side by a 3 nt linker with fixed sequence and an adaptor to facilitate conversion of the RNA into cDNA for subsequent Illumina sequencing (Fig. 1c and Supplementary Fig. 1). To prevent base pairing between adapter regions and complementary sequences in the randomized RNA regions, which would bias the available substrate pool[23,24], we annealed DNAs complementary to the adapter regions to mask the corresponding sequences (Fig. 1c).

The RNA pool was incubated in separate reactions with increasing concentrations of RbFox. Upon reaching equilibrium, samples were transferred to non-denaturing PAGE to separate RbFox-bound and -unbound substrates (Supplementary Fig. 1). Unbound substrates were isolated, converted to cDNA, and sequenced on the Illumina platform (Fig. 1d). The preparation of the cDNA libraries included unique molecular identifiers (UMIs), to identify and correct for PCR overamplification artifacts[9,23,24] (Supplementary Fig. 1). As expected, the addition of RbFox caused depletion of variants in the unbound RNA pool, compared to the control pool without RbFox (Fig. 1e).

We calculated apparent relative association constants ($K_{A,rel}$, affinities normalized to the affinity of the sequence 5′-UGCAUGU) for all substrate variants from binding reactions with increasing RbFox concentrations, as previously described[19–21]. Experiments were performed in independent duplicates, which provided highly correlated sequencing read values ($R^2 > 0.85$; Supplementary Fig. 2). Apparent affinities obtained with HiTS-Eq also correlate well with affinities measured for individual substrates by us and others[21] ($R^2 = 0.92$, Fig. 1f), indicating that the HiTS-Eq approach faithfully reflects the results of conventional biochemical assays.

The histogram of $K_{A,rel}$ values for all sequence variants represents the global affinity distribution for RbFox (Fig. 1g). Notably, the distribution is bimodal, with a large, broad peak encompassing the vast majority of the 16,384 sequence variants, and a much smaller peak with sequence variants that contain the cognate 5′-GCAUG and the closely related 5′-GCACG (Fig. 1g, small peak marked by triangle). High affinities of RbFox for 5′-GCACG containing variants are consistent with previous observations[8,11,18]. Accordingly, the sequence logo for sequence variants with the highest affinities is identical to that obtained with the RNA Bind-n-Seq (RBNS), RNACompete and HTR-SELEX approaches[8,10,11,18] (Fig. 1h). The global affinity distribution also highlights the extraordinary specificity of RbFox, compared to other RBPs (Supplementary Fig. 3).

Our data provide two insights. First, the affinity of RbFox for all sequence variants without 5′-GCA(U/C)G is lower by orders of magnitude, compared to its cognate 5-mer. To our knowledge, this extraordinary ability to discriminate against all but two closely related sequence variants has not been reported for other RBPs. Second, the bimodal affinity distribution of RbFox differs from the unimodal distributions seen for other RBPs[8,9,12,13].

### Quantitative binding models indicate two distinct binding modes

To understand the determinants of these unique features of RbFox specificity, we applied quantitative binding models, focusing on the minimal 5 nt RbFox consensus[9,12,24]. To account for variations in the register of each possible 5-mers in our RNA pool with 7 randomized nucleotides, we plotted affinity values for all 48 5-mers for each of the 1,024 5-mer variants and determined the median relative affinity for each 5-mer (Fig. 2a and Supplementary Fig. 4a). The obtained values correlate well with corresponding 5-mer affinity scores calculated from the RBNS approach[22] (Supplementary Fig. 4b), a high throughput approach designed to delineate preferred binding motifs for RBPs[8,18].

Comparison of the HiTS-Eq 5-mer and 7-mer affinities show how nucleotides flanking the core 5-mer affect RbFox binding (Supplementary Fig. 4c). Comparable affinities of 5′-GCAUG and 5′-GCACG variants depend on nucleotides flanking the core 5-mer, as previously noted[8,11,18]. Our data reveal that the impact of flanking nucleotides depends also on the 5-mer sequence. For example, a 5′-U flanking 5′-GCACG enhances the affinity, compared to a 5′-G, but for most other 5-mers, a 5′-U decreases the affinity, compared to 5′-G (Supplementary Fig. 4c).

To better understand the rules governing the binding of RbFox to the core 5-mer, we analyzed the median affinity values with a Position Weight Matrix (PWM) model (Fig. 2b), the simplest model to describe the binding of a protein to all RNA sequence variants. Although a PWM considers only the position of each nucleotide in isolation[9,12,24], the model accounts for 75% of the data variance among the 5-mers (Fig. 2b). However, the PWM bears no resemblance to the cognate 5-mer, indicating instead a preference for G at all positions (Fig. 2c). Most interestingly, the model reveals a single outlier—the cognate 5′-GCAUG sequence (Fig. 2b). These observations indicate that a PWM can

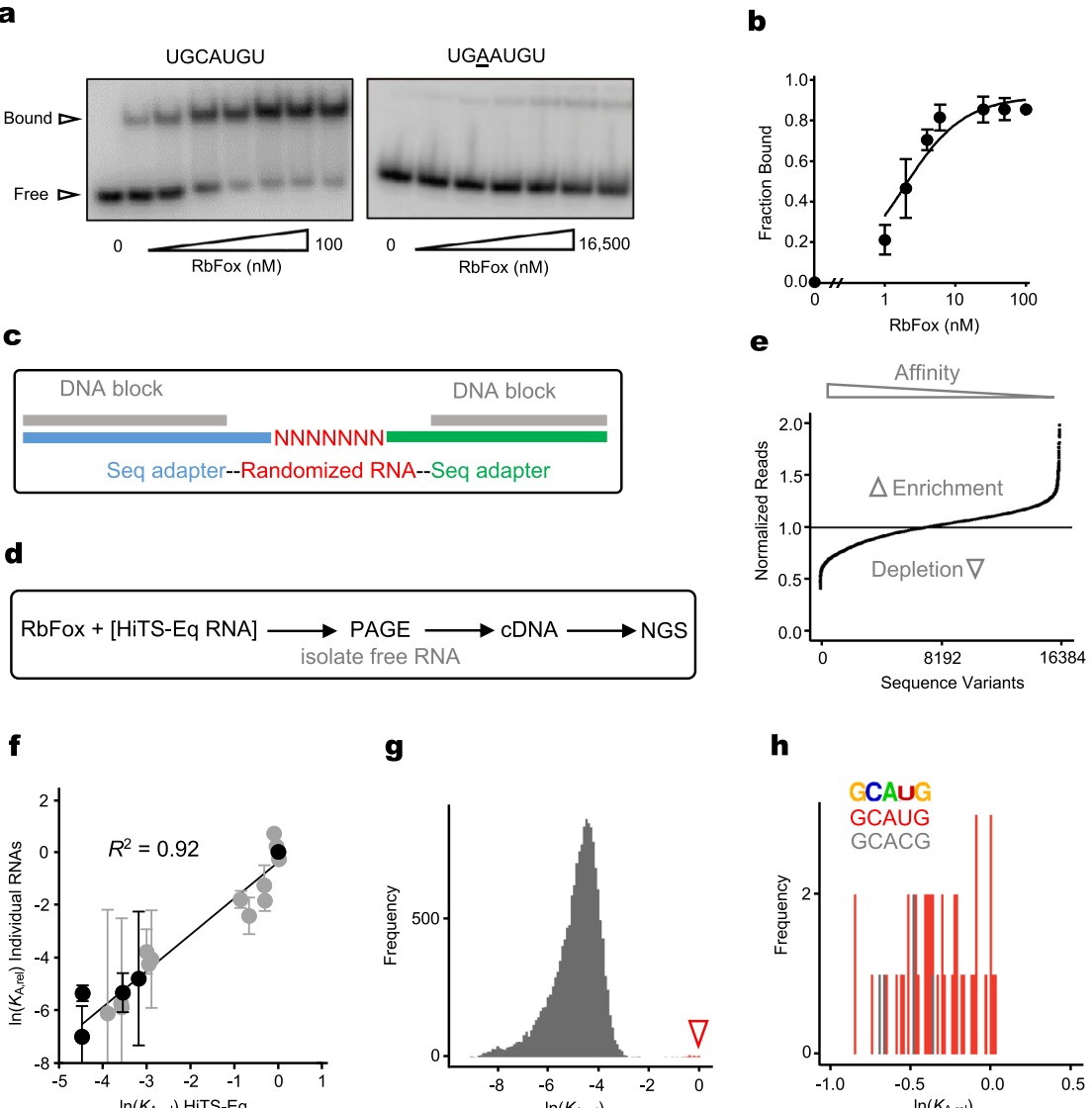

**Fig. 1 | RbFox binding to all 7-mer RNA sequence variants. a** Representative PAGE images for RbFox binding to individual RNA substrates under equilibrium conditions. (RbFox: 0, 1, 2, 4, 6, 25, 50 and 100 nM for 5′-UGCAUGU, underline marks the consensus 5-mer; 258, 516, 1,031, 2,063, 4,125, 8,250 and 16,500 nM for 5′-UGAAUGU, RNAs: 1 nM). Data were replicated independently three times with similar results. **b** Binding isotherm for 5′-UGCAUGU (data points: average of three independent experiments; error bars: one standard deviation; line: best fit to binding isotherm with $K_{1/2}^{(UGCAUGU)} = 1.6 \pm 0.3$ nM). The low level of RbFox binding to 5′-UGAAUGU (panel **a**) precludes reliable affinity determination (estimated lower limit for $K_{1/2}'^{(UGAAUGU)} > 17$ μM). **c** Design of RNA substrate pool for the HiTS-Eq measurements (detailed information: Supplementary Fig. 1). **d** Basic Scheme for the HiTS-Eq approach. **e** Depletion (normalized reads <1) and enrichment

(normalized reads > 1) of RNA sequence variants at [RbFox] = 19.74 μM. Reads are normalized to read numbers in the library without protein. **f** Relative apparent association constants ($K_{A,rel}$) for corresponding RNA variants (for variants, see Methods), measured for individual RNA and by HiTS-Eq (data points: average of three independent experiments; error bars: one standard deviation; $R^2$: correlation coefficient; black points: our measurements; gray points: values reported by Stoltz et al. [21]). **g** Affinity distribution ($K_{A,rel}$) of RbFox for all 16,384 RNA sequence variants (bin size: 100). The triangle indicates the population of high-affinity variants. **h** Distribution of high-affinity variants (bin size: 100). (Sequence motif logo: determined for 40 variants with the highest affinity; $E = 2.7e^{-77}$; red bins: variants containing 5′-GCAUG; gray bins: variants containing 5′-GCACG). Source data are provided as a Source Data file.

describe affinities for almost all sequence variants to a considerable degree, except for the cognate variant.

To examine whether the inherent limitations of the PWM cause the striking divergence between cognate and non-cognate RNAs, we applied a binding model that considers pairwise coupling (PWC) between all nucleotides[7,9,12,24]. Considering coupled contributions from multiple positions within the binding sites often improves the description of the experimental variance[2]. We analyzed the binding profiles to RbFox for 5′-GCNUG sequences (Supplementary Fig. 5), and found that substituting the G2-A4 base-pairing interaction lowers the binding affinity dramatically, highlighting the importance of couplings involving even non-adjacent nucleotides. The PWC model accounts for

90% of the data variance (Fig. 2d), an excellent result for experimental high-throughput data[9,12,13]. Yet, the cognate 5-mer remains a clear outlier, even though the PWC model highlights favorable base couplings contributions in the cognate variant (Fig. 2e, highlighted fields). PWM and PWC models for all 7-mer RNA variants, which do not consider the binding register of the minimal 5-mer, also identify variants with the cognate 5′-GCAUG as outliers (Supplementary Fig. 6). Collectively, our analyses of the RbFox-RNA interaction with quantitative binding models suggest that the bimodal affinity distribution of RbFox cannot be explained by a single binding mode. Therefore, we conclude that RbFox employs distinct binding modes—one for its cognate 5-mer and one or more other modes for non-cognate sequence variants. To

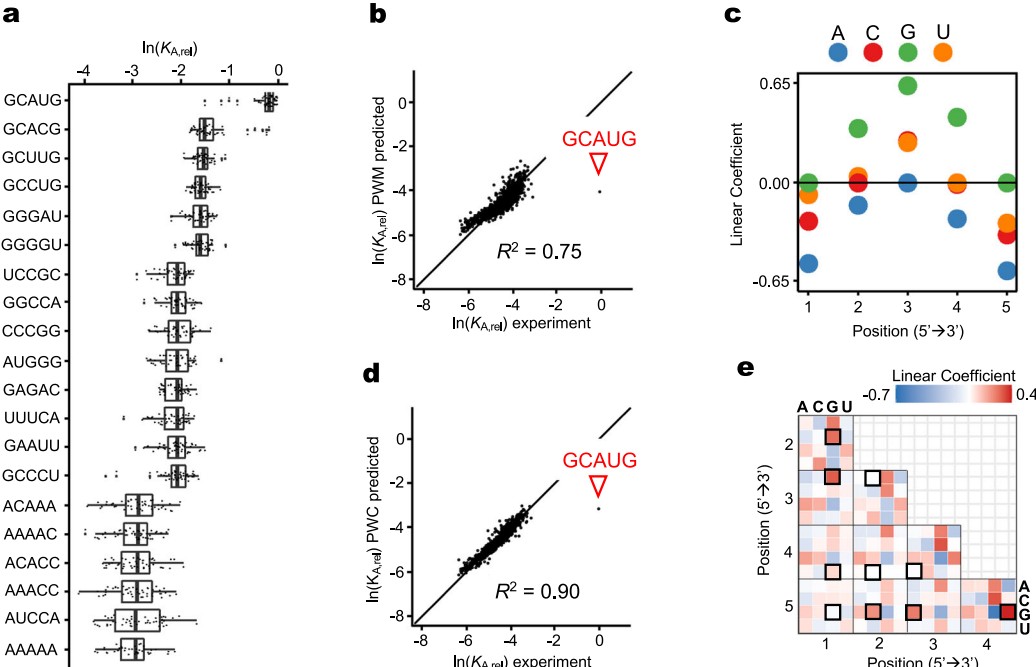

**Fig. 2 | Analysis of RbFox affinity distribution with quantitative binding models. a** Relative affinities ($K_{A,rel}$) for selected 5-mer RNA variants, indicated on the left. 48 $K_{A,rel}$ values correspond to all 5-mer with 7 randomized nucleotides (vertical line: median; box: variability through lower quartile and upper quartile; whiskers: variability outside the lower and upper quartiles). **b** Correlation between experimental $K_{A,rel}$ values for each 5-mer (median value, panel **a**) and values calculated with the Position Weight Matrix (PWM) binding model (triangle: consensus 5-mer; line: diagonal, $y = x$; $R^2$: correlation coefficient). **c** Linear coefficients for each

nucleotide position calculated with the PWM binding model (negative values: destabilization). **d** Correlation between experimental $K_{A,rel}$ values for each 5-mer (median value, panel **a**) with values calculated with the Pairwise Coupling (PWC) binding model (triangle: consensus 5-mer; line: diagonal, $y = x$; $R^2$: correlation coefficient). **e** Linear coefficients for each pairwise coupling between all nucleotides calculated with the PWC binding model (black frames: couplings in consensus 5-mer). Source data are provided as a Source Data file.

our knowledge, multiple binding modes have been reported for multi-domain RBPs, but not for a single RRM.

## Protein mutations affect the two binding modes differently

To probe the notion of two distinct binding modes, we measured the affinity distribution of an RbFox variant containing four amino acid changes (RbFox[mut], Fig. 3a). These mutations were introduced to improve binding to a pri-miRNA comprising a near-consensus sequence[20]. We hypothesized that mutations in the RRM should affect the two distinct binding modes differently. We first measured the binding of RbFox[mut] to the individual RNAs examined above with wild type (wt) RbFox (Fig. 3b). RbFox[mut] bound an RNA with the cognate 5-mer with slightly reduced affinity, compared to wt RbFox (Fig. 3b, c). However, RbFox[mut] bound to an RNA with a single base change only slightly less well than to the cognate sequence. In contrast, the binding of wt RbFox to this RNA was barely detectable (Fig. 1a, b). This observation indicates that the mutations in RbFox[mut] markedly increase the affinity for non-cognate RNAs, compared to the wt protein, while impacting binding to the cognate sequence much less.

We next examined RbFox[mut] with HiTS-Eq using the RNA pool employed above for wt RbFox (Supplementary Fig. 7). Apparent affinities obtained with HiTS-Eq correlate well with corresponding data measured for individual RNAs ($R^2 = 0.83$, Fig. 3d). However, the affinity distribution for RbFox[mut] does not display the pronounced bimodal shape observed with wt RbFox. Rather, the RbFox[mut] distribution shows a tail corresponding to high affinity variants (Fig. 3e and Supplementary Fig. 8a). In addition, the RbFox[mut] distribution narrows and is shifted towards higher relative affinities compared to the wt RbFox distribution (Fig. 3e). The global increase in relative affinities for non-cognate sequences, compared to wt RbFox,

coincides with a pronounced decrease in specificity for RbFox[mut] (Fig. 3e and Supplementary Fig. 8a); that is, the mutant protein does not discriminate against non-cognate sequences as strongly as the wt protein. In addition, the base preference of wt RbFox at position 2 of the cognate 5-mer is essentially lost in RbFox[mut] (Figs. 3f and 1h); a possible reason for the slightly lower affinity of RbFox[mut] for the cognate sequence, compared to wt RbFox (Fig. 3b). The loss of stringency at position 2 also allows RbFox[mut] to accommodate more diverse 5-mer variants in the binding mode reserved for the cognate sequence in wt RbFox.

The opposite effects of RbFox mutations on the affinities for cognate and non-cognate RNAs, strongly support the notion of distinct binding modes of RbFox. In addition, our data reveal that wt RbFox accomplishes high specificity by thermodynamically penalizing all sequences, except the cognate and near-cognate variant with a specific flanking nucleotide. RbFox[mut] is unable to impose a similar thermodynamic penalty. The affinities for non-cognate sequence variants increase relative to the cognate sequence and specificity is reduced, compared to wt RbFox. The apparent need to impose a thermodynamic penalty on all but the cognate sequence provides an additional rationale for the two distinct binding modes observed for wt RbFox (Fig. 2).

We next probed how the binding mode for the non-cognate RNA variants differed between wt and mutant RbFox. To this end, we analyzed the affinity distribution of RbFox[mut] with the PWM and the PWC models, as described for wt RbFox (Fig. 4 and Supplementary Figs. 8b, 9). The PWM model for RbFox[mut] accounted for 35% of the data variance and differed from the PWM for wt RbFox, most notably by a preference for U, instead of G (Supplementary Fig. 9c). The PWC model accounted for 66% of the data variance for RbFox[mut] (Fig. 4a, b) and revealed numerous differences with wt RbFox (Fig. 4c), highlighting a

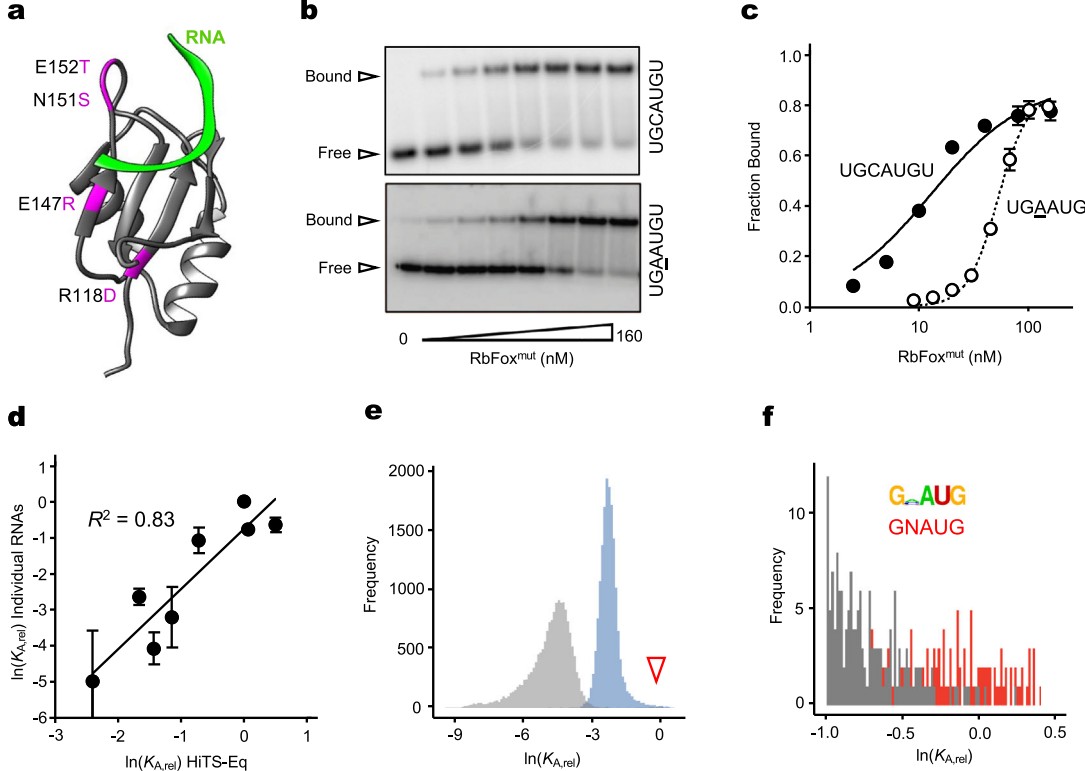

**Fig. 3 | Impact of RbFox mutations on the affinity distribution. a** Locations of the mutations in RbFox[mut] in the RRM[20], highlighted in purple (green band: RNA 5′-UGCAUGU[19]). **b** Representative PAGE images for RbFox[mut] equilibrium binding to individual RNA substrates (sequences on the right; RbFox[mut]: 0, 2.5, 5, 10, 20, 40, 80 and 160 nM for both substrates; RNAs: 1 nM). The experiments were repeated three times with similar results. **c** Equilibrium binding isotherm for RbFox[mut] with the substrates shown in panel **b** (data points: average of three independent experiments; error bars: one standard deviation; lines: best fit to binding isotherm; $K_{1/2}^{(UGCAUGU)} = 14.3 \pm 2.5$ nM, $K_{1/2}^{(UGAAUGU)} = 35.3 \pm 11.8$ nM). **d** Relative apparent association constants ($K_{A,rel}$) for corresponding RNA variants (for sequences, see

Methods), measured for individual RNAs and by HiTS-Eq (data points: average of three independent experiments; error bars: one standard deviation; $R^2$: correlation coefficient). **e** Affinity distribution ($K_{A,rel}$) of RbFox[mut] for all 7-mer RNA sequence variants (blue) (bin size: 100). For reference, the affinity distribution of wild type RbFox (gray) is plotted as well (triangle: population of high affinity variants for wt RbFox). **f** Distribution of high affinity variants for RbFox[mut] (bin size: 100). Sequence motif logo was determined for 40 variants with the highest affinity; $E = 5.6e^{-57}$; red bins: variants containing 5′-GNAUG; gray bins: variants that differ from 5′-GNAUG. Source data are provided as a Source Data file.

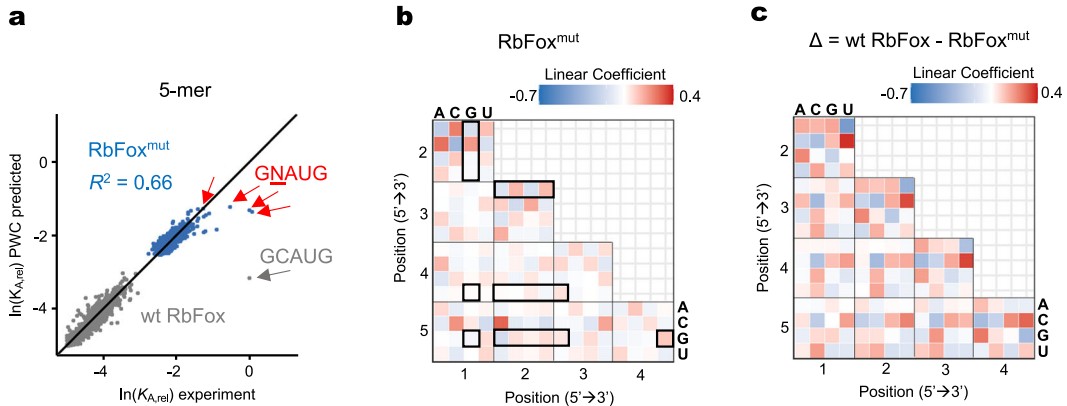

**Fig. 4 | Analysis of RbFox[mut] affinity distribution with quantitative binding models. a** Correlation between experimental $K_{A,rel}$ values for RbFox[mut] for each 5-mer (median values, Supplementary Fig. 8) with values calculated with the PWC binding model (blue dots: RbFox[mut]; red arrow: consensus 5-mer 5′-GNAUG; line: diagonal, $y = x$; $R^2$: correlation coefficient). The plot for wt RbFox (gray dots) is plotted as reference. **b** Linear coefficients for each pairwise coupling between all

nucleotides calculated with the PWC binding model; (black frames: couplings in consensus 5-mer, negative values: destabilization). **c** Differences between linear coefficients for PWC binding model for wt RbFox, compared to RbFox[mut]; (positive values: increase in wt RbFox, compared to RbFox[mut], negative values: decrease in wt RbFox, compared to RbFox[mut]). Source data are provided as a Source Data file.

significant specificity shift by only 4 mutations of RbFox. Collectively, these data indicate substantial differences in the binding mode for the non-cognate variants between RbFox[mut] and wt protein, thereby supporting the notion of distinct binding modes.

## Cognate and non-cognate RNAs induce distinct structures in the RbFox RRM

The existence of two distinct binding modes of RbFox raised the question of whether these binding modes also differ structurally. To

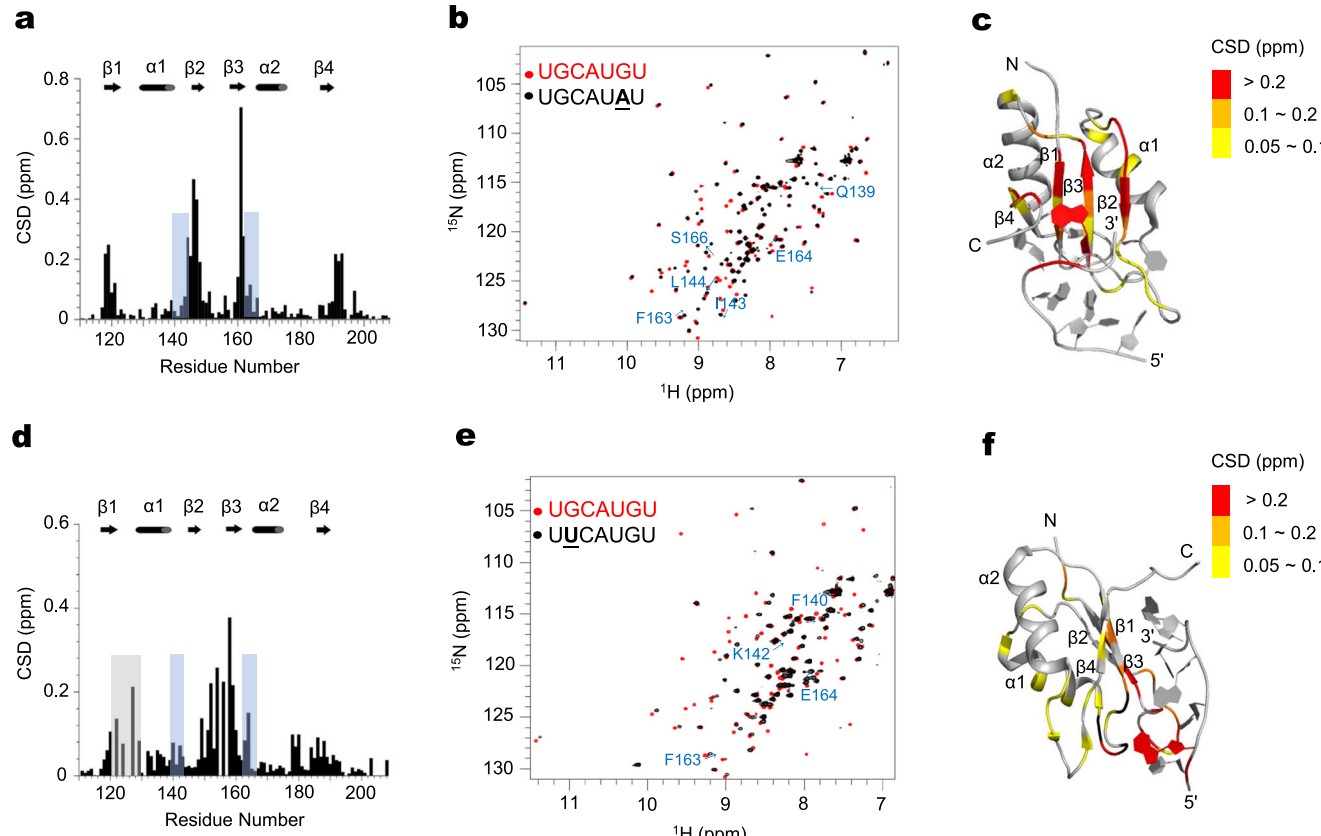

**Fig. 5 | NMR analysis of the interaction of RbFox RRM with its consensus RNA and two RNA variants. a** Chemical shift difference (CSD) between RbFox RRM bound to its cognate RNA 5′-UGCAUGU and RNA1 5′-UGCAUAU. Regions with significant CSD while distant from the RNA-binding site are highlighted in blue. **b** Superposition of ¹H-¹⁵N HSQC spectra of RbFox RRM complexed with the two RNAs (red: 5′-UGCAUGU; black: 5′-UGCAUAU). Residues with significant CSDs distant from the RNA-binding site are labeled in blue. **c** Mapping of the CSD in panel **a** onto the structure of RbFox bound to its cognate RNA (pdb #2ERR [https://doi.org/10.2210/pdb2err/pdb]) (red: CSD > 0.2 ppm; orange: 0.1 ppm<CSD< 0.2 ppm; yellow: 0.05 ppm<CSD< 0.1 ppm). **d** CSD between RbFox RRM bound to its cognate

RNA 5′-UGCAUGU and RNA2 5′-UUCAUGU. Residues broaden out due to intermediate chemical exchange are highlighted in gray, and residues with significant CSD while distant from the RNA-binding site are highlighted in blue. **e** Superposition of ¹H-¹⁵N HSQC spectra of RbFox RRM complexed with the two RNAs in panel **d** (red: 5′-UGCAUGU; black: 5′-UUCAUGU). Residues with significant CSDs distant from the RNA-binding site are labeled in blue. **f** Mapping of the CSD in panel **d** onto the structure of RbFox bound to its cognate RNA (pdb #2ERR [https://doi.org/10.2210/pdb2err/pdb]) (same color scheme as in panel **c**). Source data are provided as a Source Data file.

examine this possibility, we performed NMR studies with a 7-nt cognate (5′-UGCAUGU) and two non-cognate RNAs, which differ from the cognate variant at a single nucleotide (5′-UGCAU**A**U) or (5′-U**U**CAUGU), designated as RNA1 and RNA2. RNA1 and RNA2 both represent examples for non-cognate sequence binding modes, and rank in the 17.4 and 37.6 percentile, respectively, among the 16,384 sequence variants. We titrated each RNA into the wt RbFox RRM until saturation, while monitoring ¹H-¹⁵N HSQC (Supplementary Fig. 10). We observed slow-exchange binding kinetics with the cognate RNA and intermediate-exchange binding kinetics with the non-cognate variants (Supplementary Fig. 10), consistent with the nanomolar affinity of the cognate RNA and the markedly lower association constants of the non-cognate variants observed in the HiTS-Eq data (Fig. 2).

Next we collected 3D NMR spectra at saturating RNA concentrations and calculated the chemical shift difference (CSD) between the RbFox complexes bound to the consensus RNA and to RNA1 and RNA2 (Fig. 5a, d). For RNA1, mapping of these differences onto the structure of wt RbFox identified changes in four regions of the protein: β1(R118-V121), β2 and the following loop (I143-E152), β3 (G159-E164) and the C-terminal tail (A191-A193) (Fig. 5c). In the structure of RbFox RRM complexed with its cognate RNA, the AUG (nucleotides 4 to 6) element is bound in a canonical manner through π-π stacking with the β-sheet surface, while UGC (nucleotides 1 to 3) is recognized by loop residues[19]. Accordingly, similar chemical shifts for the region of RbFox interacting

with UGC were observed in the two complexes, indicating that the UGC nucleotides in both structures maintain the same contacts within the RNA-binding cleft of RbFox. In contrast, the G- > A substitution at position 6 causes marked changes in the β-sheet region that binds to the AUG element (0.2–0.7 ppm, Fig. 5a). We also observed significant chemical shift changes (>0.1 ppm) in the α1/β2 and β3/α2 loops (highlighted and labeled in blue in Fig. 5a, b), which are distant from the RNA-binding surface (Fig. 5c). For RNA2, where G2 is mutated to U, we observed a similar pattern of CSDs, not only in the direct RNA-binding cleft, but also on the distal α1/β2 and β3/α2 loops (Fig. 5d–f). Mapping of these differences onto the structure of wt RbFox protein revealed three major regions directly contacting G2 and its base paring partner A4: β1 and its following loop (L119-R129), β3 and its preceding loop (I149-F160), and β4 and its preceding loop (G178-V188). Compared to the G6- > A mutation, the distal α1/β2 and β3/α2 loops of RbFox are affected by the G2- > U mutation to a similar extent, although the overall changes induced by the G2- > U mutation are much smaller.

To gain detailed insight into the structural differences between the RbFox RRM structures with cognate and non-cognate RNAs, we determined the structure of the RbFox RRM with the RNA1 substrate. The ¹H-¹⁵N HSQC and NOESY spectra changed during data collection, suggesting multiple binding orientations that are most likely due to the weak binding of the non-cognate RNA. These characteristics made

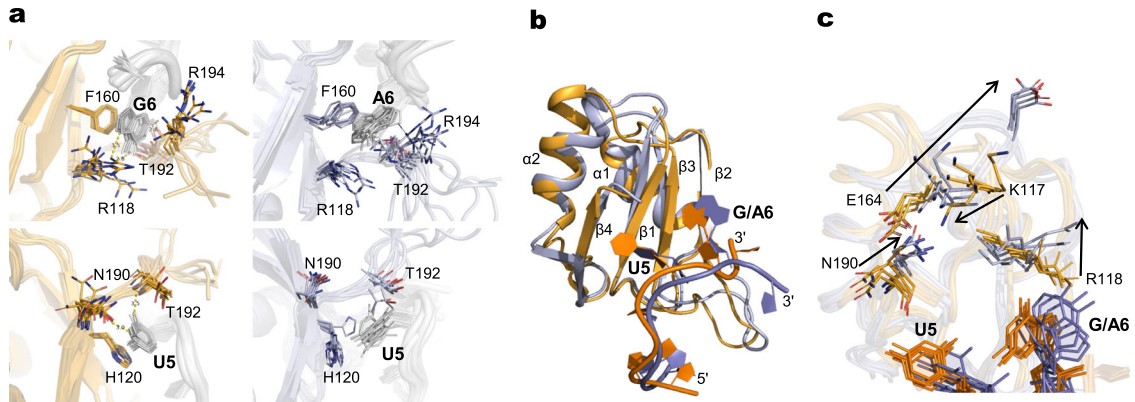

**Fig. 6 | Comparison of RbFox RRM structures with 5'-UGCAUGU and 5'-UGCAUAU. a** Recognition of G6/A6 (upper panel) and U5 (lower panel) by RbFox RRM (left: 5'-UGCAUGU; right: 5'-UGCAUAU). Side chains of key residues in the interactions (sticks) are labeled. **b** Comparison of the two RbFox-RNA structures (blue: complex with 5'-UGCAUGU; orange: complex with 5'-UGCAUAU). **c** Structural path for the long-range conformational rearrangement of RbFox upon binding to the non-cognate RNA (color scheme as in panel **b**; arrows: reorientation of the side chains involved in the path).

unambiguous NOE identification difficult. To overcome this problem, when collecting spectra for the protein, we oversaturated the protein with RNA and changed the sample frequently during data collection to ensure RbFox was consistently in the fully bound conformation. For the RNA, we used some NOEs for the GCA sequence from our previous NMR dataset within the pre-miR20b stem loop in complex with RbFox[25]. Since chemical shifts are very similar for both proteins and RNAs, we obtained a dataset that converged to a reliable structural ensemble (Supplementary Table 1). RNA2 binds even more weakly, as demonstrated by highly inhomogeneous peak intensities (Fig. 5e) and the loss of multiple HN signals in the $^1$H-$^{15}$N HSQC (highlighted in gray in Fig. 5d); because of the poor spectral quality resulting from reduced affinity, the structure of RbFox bound to RNA2 could not be determined.

A 2.0 Å rmsd difference was observed for RbFox in complex with the cognate and non-cognate RNAs, with large differences in protein regions distant from the RNA-binding site as well (Fig. 6). In the NMR structure with the cognate sequence, G6 forms the most extensive contacts with the protein (Fig. 6a, left panel). In our structure with the non-cognate RNA, the G6->A substitution abolishes the hydrogen-bonding interactions with R118 and T192. Our structure also reveals increased dynamics for A6 and stacking interactions between the aromatic chain of F160 and the aliphatic chain of R194 (Fig. 6a, right panel). The hydrogen bonding network of U5 is also rearranged due to loosened interactions between A6 and the protein. With the non-cognate RNA, the stacking between the imidazole ring of H120 and U5 is lost or weakened, because the side chain of H120 is flexible, as revealed by line broadening in the NMR spectra. The hydrogen-bonding interactions with N190 are completely lost, and those with T192 can only be observed for less than half of the models in the structural ensemble (Fig. 6a). For a large majority of the converged structures, the aromatic ring of F160 tilts to stack with the base of A6. The RNA-binding strands β1 and β3 float away from the RNA, due to the loss of close contacts with A6 and U5, and the β2 β3 loop reorients to better accommodate the non-cognate RNA (Fig. 6b).

Collectively, these structural features indicate that RbFox employs structurally distinct binding modes for the cognate and the non-cognate RNA, sacrificing the binding energy of multiple hydrogen bonds and π−π stacking interactions in order to accommodate a non-cognate sequence. These observations provide structural evidence for the thermodynamic penalty model.

Equally intriguingly, the non-cognate RNA induces a distinct long-range conformational rearrangement in RbFox, which remodels the surface on the protein distant from the RNA-binding site (Fig. 6b, c). The side chain of R118 rotates almost 90 degrees to interact with the Watson-Crick face of A6, thereby losing a hydrogen bond with the Hoogsteen face of G6, that can only be formed with the cognate RNA (Fig. 6c). The rotation of R118 leads to a reorientation of the side chain of K117. These rearrangements, together with the reoriented side chain of N190 and flipping out of the side chain of E164, are necessary to relieve steric clashes (Fig. 6c). This domino-like effect causes the movement of the N-terminal loop, β3-α2 loop and the α2 helix, propagating the conformational change to peripheral structural elements α1 and β2 and the connecting loop. A new surface is created by the above rearrangements, exemplified by the negative-charged residue E164 flipping out and altering the protein surface electrostatics (Supplementary Fig. 11). Thus, a single nucleotide change in the RNA triggers not just binding mode alterations at the RNA-binding site, as would be expected, but also at a distal surface, where other factors potentially interact with RbFox to modulate its biological activities[26].

We carried out an NMR analysis to investigate whether wt RbFox's divergent binding mode is structurally lost in the mutant. We used two RNAs, a 7-nt cognate RNA 5'-UGAAUGU and a non-cognate RNA 5'-UGAAUCU, to titrate into RbFox$^{mut}$ and collected $^1$H-$^{15}$N HSQC spectra, as was done for wt RbFox (Supplementary Fig. 12a, b). In contrast to wt RbFox, we observe slow-exchange on the chemical shift time scale with both RNAs, indicating both cognate and non-cognate RNAs bind to RbFox$^{mut}$ with strong affinity, better than μM. Furthermore, taking advantage of the tight binding, we were able to collect high quality 3D NMR spectra to complete the backbone HN assignments for both RbFox$^{mut}$ complexes and compared the CSDs between them. Compared to wt RbFox, fewer residues in RbFox$^{mut}$ protein are affected by the single-nucleotide mutation and smaller CSDs are observed, suggesting the chemical shift changes of RbFox$^{mut}$ originate mainly from differences in the chemical environment caused by the nucleotide change, but not by a structural change (Supplementary Fig. 12c–e). CSDs distal from the RNA-binding surface induced by the G6->A or G2->U mutations, as observed with wt RbFox, were not observed for the G6->C mutation with RbFox$^{mut}$. Taken together, these results provide strong evidence that the conformational divergence induced by RNA substrates observed with wt RbFox is weakened in RbFox$^{mut}$. This observation supports our notion that a pronounced bimodal binding mode is a unique characteristic of wt RbFox.

## Discussion

Specificity variations are critical determinants of RBP function[1,2,5,8,10,11,14]. Defining the molecular mechanisms that determine RBP specificity is thus necessary for understanding RNA biology. Here, we have shown that a highly specific RNA-binding module, the RNA recognition motif of the RbFox proteins, accomplishes extraordinary

sequence specificity through a previously unknown mechanism, the use of two structurally distinct binding modes. One binding mode exclusively accommodates cognate and near-cognate RNAs with high affinity, while the other binding mode accommodates all other RNAs, but with markedly reduced affinity.

The notion of distinct binding modes for the RRM of RbFox is based on three converging, independent lines of evidence. First, the affinity distribution of RbFox for all 16,384 7-mer RNA sequence variants is bimodal (Fig. 1). In contrast, affinity distributions for other RBPs are unimodal[8,9,12], reflecting the gradual incremental effects of protein sequence variations on RNA affinity[22]. A single binding mode can be modeled by either a position weight matrix or more complex models considering functional coupling between bases[24,27] (e.g., pairwise interaction matrix). However, the bimodal affinity distribution of RbFox cannot be described by a single binding model.

Second, mutations in RbFox affect the two binding modes differently. The mutations only slightly decrease the affinity for the cognate RNA variant, but markedly increase affinity for all other variants, thereby reducing overall specificity (Fig. 3). Related observations were reported for the RBP FBF-2, a member of the PUF-protein family, where mutations in the RNA-binding site increased accommodation for multiple sequence variants[28].

Third, structural differences between complexes of RbFox with the cognate and a non-cognate sequence are observed (Figs. 5, 6). Beyond expected differences at the RNA-protein interface, we also identify pronounced differences in regions distant from the RNA-binding site. In other RBPs for which structures with different RNA variants have been reported, structural variations are generally concentrated at the RNA-binding sites[29–31], even where relatively large changes are introduced in the protein[30,32]. Our data with RbFox reveal instead that structural rearrangements in the RRM can be communicated to a distant surface. While this is not unprecedented, it is also intriguing that the conformational change depends on whether a cognate or non-cognate RNA is bound to the protein. The structural analysis suggests that the different binding modes are accomplished in a switch-like manner: the cognate variant induces one structure while non-cognate variants another. It is possible that additional, but presumably smaller structural differences occur within the non-cognate binding mode, similar to the differences that have been observed for other RBPs bound to multiple different RNAs[29–31,33]. Structural and functionally, the propagation of the conformational rearrangements to sites distant from the RNA interface reveals that RNA can act as an allosteric effector, potentially regulating the binding of other proteins to RbFox, and altering the composition of multi-component RNP complexes, including the Large Assembly of Splicing Regulators (LASR)[34].

The use of distinct binding modes by a single RRM to achieve high sequence specificity is an interesting concept for single protein domains. Distinct binding modes have been reported for proteins with multiple domains, reflecting the differential contributions of each RRM for binding to complex multimodal RNA sites; an important and well-understood advantage of their modular structure[31,35–37]. However, these scenarios differ fundamentally from the multiple binding modes of a single RNA-binding module shown here. The two binding modes of the RbFox RRM enable exceptionally high sequence selectivity by enabling the imposition of a high thermodynamic penalty on the non-cognate sequences. Mutations in the RRM diminish the ability to maintain the high thermodynamic penalty because the affinity for non-cognate variants increases and specificity decreases (Fig. 3).

We speculate that the two binding modes evolved to allow sharp discrimination between small sequence variations within a single RRM, which might be impossible to accomplish within a single binding mode, as reflected by the generally relaxed specificity of many RRMs[38]. To accomplish an RbFox-like difference between cognate and near-cognate sequences within a single binding mode would require a range in the affinity distribution that might not be physically possible, biologically desirable, or both. Since RNA-protein association is universally limited by diffusion, affinities in the sub-nanomolar range or higher, are necessarily associated with long lifetimes of the RNA-protein complex[39], which might be incompatible with RBP functions in pre-mRNA splicing regulation, where remodeling of RNA-protein interactions occurs on the scale of minutes or faster[1,2,39]. Low affinities for RNA-protein interactions are limited by the electrostatic properties of RNA, which promote "non-specific" interactions with proteins that are usually in the low to mid micromolar range[1,2,39]. Therefore, we propose that the dual binding modes for RbFox evolved to overcome the physical limits of sequence discrimination imposed by a single binding mode.

It remains an open question why RbFox requires higher specificity than other RBPs. Recent data show that RbFox binds and promotes biological effects also at near-cognate RNA sites in cells, but only at increased cellular RbFox concentrations[22]. The extraordinary specificity of RbFox might thus be necessary for high fidelity binding to the cognate sequence at low RbFox concentrations, which is likely critical for accurate control of splicing networks in brain, heart, and muscle, and during embryonic development[16]. In addition, the different conformations of RbFox on cognate and non-cognate sequences described herein, might promote the assembly of distinct regulatory complexes, which allow RbFox to transmit even minute changes in RNA sequence to potential binding partners. The structural differences between the two binding modes thus add a previously unappreciated layer of regulation to RBP biology.

## Methods

### Protein expression and purification
Expression and purification of RbFox RRM and RbFox[mut] were performed as previously described[20,25]. Briefly, the recombinant plasmid was transformed into BL21 (DE3) competent cells. Transformants were grown in LB media at 37 °C to OD$_{600nm}$ = 0.6. The recombinant proteins were overexpressed in *E. coli* at 18 °C overnight upon induction of 0.5 mM IPTG. The cells were harvested by centrifugation and resuspended in lysis buffer (20 mM Tris at pH 8.0, 150 mM NaCl and 1 mM DTT) and lysed by sonication at 4 °C. The crude extracts were centrifuged at 27,000 × g for 60 min to remove the cell debris. To purify the recombinant proteins, the supernatant was applied onto a HisTrap column (GE Healthcare) and a Heparin column (GE Healthcare) in succession. Protein concentrations were measured by UV absorbance at 280 nm and confirmed by the Bradford assay.

Protein preparation for NMR experiments required complete removal of traces of RNase contaminants, due to the long incubation and weak binding of wt RbFox to the non-cognate RNA. To accomplish this, RbFox was double tagged with both Hisx6 and GST at its N-terminus and purified with HisTrap and GSTrap columns in succession, followed by a Heparin column to remove residual non-specifically bound RNAs. Both affinity tags were then cleaved with TEV protease and removed by Nickel affinity chromatography. The purified protein eluent was concentrated and loaded onto a Superdex 75 10/300 GL column equilibrated with NMR buffer (10 mM sodium phosphate at pH 6.0 with 30 mM NaCl). Aliquoted protein samples were flash-frozen for storage at −80 °C.

### RNA substrates for equilibrium binding and HiTS-Eq measurements
RNA substrates were purchased from Dharmacon (Lafayette, CO) and Sigma-Aldrich (St. Louis, MO). RNA substrates were 5′ end labeled using γ$^{32}$P-ATP and T4 Polynucleotide Kinase (NEB, MA)[40]. Uniformly radiolabeled RNA substrates were purified with 20% denaturing PAGE (acrylamide/bis 19:1, 7 M urea, 0.5 × TBE) and concentrations were quantified by scintillation counter.

**Table 1 | Substrates for HiTS-Eq measurements**

| Name | Sequence (5'→ 3') |
|---|---|
| RNA randomized | GGGAGACCGGAAUUCAGAUUGUCCNNNNNNNUUAAAUCCCGUCGUAGCCACCA |
| DNA 1 | CAATCTGAATTCCGGTCTCCC |
| DNA 2 | TGGTGGCTACGACGGGATT |

**Table 2 | Substrates for affinity measurements with individual RNAs**

| Measurement Series | Sequence (5'→ 3') |
|---|---|
| Own measurements[a]; wt RbFox | UGCAUGU |
| | UGCGUGU [a] |
| | UGCAAGU [a] |
| | UACAUGU [a] |
| | UGCAUAU [a] |
| Data from[b,21]; wt RbFox | UGCAUGA [b] |
| | UGCAUGC [b] |
| | UGCAUGU [b] |
| | UGCAUGG [b] |
| | CGCAUGU [b] |
| | AGCAUGU [b] |
| | UGCACGU [b] |
| | GGAAUGU [b] |
| | GGGUUGU [b] |
| | GGCUUGU [b] |
| | UGCUUGU [b] |
| | CGGUUGU [b] |
| | UGCCUGU [b] |
| Own measurements[a]; RbFox[mut] | UGCAUGU |
| | UGGAAUG |
| | CGAAUGU [a] |
| | GCAUGUC [a] |
| | GCAUGGG [a] |
| | GGCCAUG [a] |
| | UGCAUAU [a] |
| | UACAUGU [a] |

[a]RNAs used in competition experiments.
[b]Measurements were conducted by surface plasmon resonance (BIACORE)[21].

For the HiTS-Eq randomized RNA pool, 5' end radiolabeled randomized RNA substrates containing a central segment of 7 randomized (NNNNNNN) nucleotides flanked by illumina sequencing primers (underlined sequences) were annealed with two complementary DNA oligonucleotides (underlined sequences) (Table 1). The RNA–DNA complexes were purified by 15% non-denaturing PAGE (acrylamide/bis 19:1, 0.5 × TBE) and concentrations were quantified by scintillation counting[40].

**Equilibrium binding with individual RNAs**
RNA-protein binding reactions for wt RbFox with consensus RNA and for RbFox[mut] with all RNAs were performed at 30 °C (10 mM Tris, 150 mM NaCl, 3.4 mM EDTA, 0.001% NP-40, pH 7.5, 5' end radiolabeled RNAs: 1 nM) and protein concentrations as indicated in the Figures (for sequences see Table 2). RNA and protein were incubated for 10 min. Longer incubation times did not change the observed fractions of bound RNA. Samples were then loaded on 8% non-denaturing PAGE. Gels were dried, and radioactivity in bound and free RNA was quantified using Phosphorimager (GE) and ImageQuant 5.2 software (GE Healthcare, IL). Plots of the fraction bound RNA vs. protein concentrations were fitted against the quadratic binding equation using KaleidaGraph (v3.52).

$$\text{Fraction Bound} = A \times \frac{(K_{1/2} + R_0 + P_0) - \sqrt{\{(K_{1/2} + R_0 + P_0)^2 - 4 \times R_0 \times P_0\}}}{2 \times R_0} \quad (1)$$

($A$: reaction amplitude, $K_{1/2}$: apparent dissociation constant, $R_0$: RNA concentration, $P_0$: protein concentration).

Affinities for wt RbFox for non-consensus RNAs were measured by competition with consensus RNA, due to low affinities of wt RbFox for these RNA variants. Consensus RNA (final concentration 1 nM) was incubated with RbFox for 10 min. Competitor RNAs at increasing concentrations were added and incubated for 10 min. Longer incubation times did not change observed fractions of bound and free RNA. Samples were loaded on 8% non-denaturing PAGE, as above. Gels were dried and radioactivity in bound and free bands were quantified using a Phosphorimager (GE) and ImageQuant 5.2 software (GE Healthcare, IL). Plots of the fraction bound RNA vs. chase RNA concentrations were fitted against the binding equation for competitive inhibition using KaleidaGraph[41].

$$\text{Fraction Bound} = A \times \frac{R_1}{R_1 + [K_{1/2}]_1 \times \left(1 + \frac{R_2}{[K_{1/2}]_2}\right)} \quad (2)$$

($A$: reaction amplitude, $R_1$: Substrate concentration, $R_2$: Competitor concentration, $[K_{1/2}]_1$: apparent dissociation constant for substrate, $[K_{1/2}]_2$: apparent dissociation constant for competitor)

**HiTS-Eq measurements and sequencing library preparation**
HiTS-Eq reactions (50 μL) were performed with 1 nM of 5' end radiolabeled RNA pool with increasing protein concentrations, as indicated in the figures. RNA and protein were equilibrated for 10 min. Longer incubation times did not change the fraction of bound and free RNA. Bound and free RNAs were separated by 8% non-denaturing PAGE. The gel was exposed for 30 min to autoradiography film, free RNA species were located, cut out, and eluted from the gel and RNA was extracted and recovered as previously described[40].

The HiTS-Eq libraries for Illumina next generation sequencing (NGS) were generated from the eluted RNA as previously described[9,12,13,23]. The RNA was reverse transcribed into cDNA with Superscript III Reverse Transcriptase (Invitrogen, CA). The cDNA was PCR amplified into HiTS-Eq libraries using index barcode primers (Table 3). The HiTS-Eq DNA libraries were purified with 8% non-denaturing PAGE. DNAs with 137 bp were cut out and extracted as previously described[9,12,23]. DNA concentration and quality were analyzed using a DNA Bioanalyzer chip (Agilent, CA). Subsequently, the HiTS-Eq libraries for different protein concentrations were pooled (equimolar) and sequenced using 50 bp single-end reads on Illumina HiSeq 2500.

**Processing of Illumina sequencing data**
Raw sequencing reads were quality checked with FastQC v0.11.5 and de-multiplexed based on their corresponding index barcode primers using Novocraft v3.0.8 (http://novocraft.com) or the Fastx-Toolkit v0.0.13 (http://hannonlab.cshl.edu/fastx_toolkit/). The sequencing reads were then aligned to the sequence of nucleotides 6–29 of the single strand randomized substrate, allowing one mismatch but no gaps using Perl scripts. The counts of the individual 7-mer sequences and two degenerative random nucleotides in the index barcode primers were added and exported into excel tables using Perl scripts (https://github.com/hsuanchunlin/HiTS-EQ, https://github.com/xxy103/HiTS-EQ).

**Table 3 | Primers for HiTS-Eq libraries**

| Forward index (5'→ 3') | Indexed reverse library PCR primer (5'→ 3') |
|---|---|
| RT primer | CAAGCAGAAGACGGCATACGATGGTGGCTACGACGGGAT |
| Index 1 | AATGATACGGCGACCACCGAGATCTACACTCTTTCCCTACACGACGCTCTTCCGATCT**NN**ATCGGGAGACCGGAATTCAGATTG |
| Index 2 | AATGATACGGCGACCACCGAGATCTACACTCTTTCCCTACACGACGCTCTTCCGATCT**NN**GATGGGAGACCGGAATTCAGATTG |
| Index 3 | AATGATACGGCGACCACCGAGATCTACACTCTTTCCCTACACGACGCTCTTCCGATCT**NN**CGAGGGAGACCGGAATTCAGATTG |
| Index 4 | AATGATACGGCGACCACCGAGATCTACACTCTTTCCCTACACGACGCTCTTCCGATCT**NN**TCCGGGAGACCGGAATTCAGATTG |
| Index 5 | AATGATACGGCGACCACCGAGATCTACACTCTTTCCCTACACGACGCTCTTCCGATCT**NN**CACGGGAGACCGGAATTCAGATTG |
| Index 6 | AATGATACGGCGACCACCGAGATCTACACTCTTTCCCTACACGACGCTCTTCCGATCT**NN**TGTGGGAGACCGGAATTCAGATTG |
| Index 7 | AATGATACGGCGACCACCGAGATCTACACTCTTTCCCTACACGACGCTCTTCCGATCT**NN**ACTGGGAGACCGGAATTCAGATTG |
| Index 8 | AATGATACGGCGACCACCGAGATCTACACTCTTTCCCTACACGACGCTCTTCCGATCT**NN**GTAGGGAGACCGGAATTCAGATTG |

Index barcode forward primer (**NN**, degenerate nucleotides)

## Calculation of binding constants from HiTS-Eq data

Relative equilibrium constants ($K_{A,rel}$) were calculated from concentration dependent changes of the 7-mer sequence variants analogous to the procedure described previously[9,12,23]. The ratios of the two RNA substrates $S_1$ and $S_2$ at a given protein concentration ($E$) was described using a competitive binding scheme according to: $S_{1,0}$ and $S_{2,0}$ are initial concentrations of the RNA substrates $S_1$ and $S_2$ at HiTS-Eq control reaction, and equilibrium constants are described as $K_1$ and $K_2$.

$$\frac{S_1}{S_2} = \frac{S_{1,0}}{S_{2,0}} \frac{1+\frac{E}{K_2}}{1+\frac{E}{K_1}} \qquad (3)$$

($S_{1,0}$ and $S_{2,0}$: initial concentrations of the RNA substrates $S_1$ and $S_2$ at the HiTS-Eq control reaction, $K_1$ and $K_2$: equilibrium constants)

The association constant ($K_2$) for $S_2$ is:

$$K_2 = \frac{E}{\left(\left(\frac{S_1}{S_2}\right)\left(\frac{S_{1,0}}{S_{2,0}}\right)\left(1+\frac{E}{K_1}\right)\right) - 1} \qquad (4)$$

The relative association constant for a given RNA variant ($K_{2,rel}$) is the value for the given RNA variant ($S_2$), normalized by the value for substrate $S_1$ [UGCAUGU], according to:

$$K_{2,rel} = \frac{E}{\left(\left(\frac{S_1}{S_2}\right)\left(\frac{S_{1,0}}{S_{2,0}}\right)(1+E)\right) - 1} \qquad (5)$$

$K_{A,rel}$ values for each 7-mer RNA variants were calculated using the sequencing read counts over protein concentrations. The $K_{A,rel}$ values for 5-mer RNA variants were determined by splitting each 7-mer substrates into 3 non-overlapping 5-mer substrates and averaging their $K_{A,rel}$ values.

## Quantitative binding models

The relative association constants were fitted with the Position Weight Matrix (PWM) model as previously described[9,12,23]. For each sequence variant, the predicted association constant ($K_A$) value is determined by a set of linear coefficients at individual nucleotide positions, according to:

$$\ln(K_{A,rel}) \sim \sum_{i=1}^{n} \alpha_i A_i + \beta_i C_i + \gamma_i G_i + \delta_i U_i \qquad (6)$$

The numbers for $N_i$ (N = A, C, G, U) are based on the nucleotide identity at position $i$ ($i = 1 - n$, $n$ refers to the length of the sequence motif). $N_i$ was assigned to value 1 for matched nucleotide at position $i$, and 0 otherwise. The cognate sequence motif UGCAUGU or GCAUG for 5-mers were used as baseline and excluded from the linear regression for the PWM models. Linear coefficients for the PWM model correspond to the parameters $\alpha_i$, $\beta_i$, $\gamma_i$ and $\delta_i$, $i = 1 - n$.

The Pairwise Coupling (PWC) model considers all pairwise interactions between two nucleotides[9,12]. Interaction coefficients between individual nucleotide positions were added to the PWM model to fit the $K_{A,rel}$ data from HiTS-Eq in equation, as previously described[9,12,23]. Interaction coefficient values ($I_n$) with a $T$ value larger than 3.5 were considered statistically significant. Obtained interaction coefficients values were plotted as heatmaps.

$$\ln(K_{A,rel}) \sim \sum_{i=1}^{n} (\alpha_i A_i + \beta_i C_i + \gamma_i G_i + \delta_i U_i) + \sum_{i=1}^{n} \alpha_n I_n \qquad (7)$$

## NMR sample preparation and data collection

RNAs for NMR measurements were purchased from Integrated DNA Technologies. Oligonucleotides were dissolved in RNase-free water, desalted using a PD MiniTrap G-10 and re-suspended in NMR buffer after overnight lyophilization. $^{13}$C, $^{15}$N labeled RRM of RbFox (residues 109-208) was prepared in minimal M9 medium supplemented with 1 g/L $^{15}$NH$_4$Cl and 2 g/L $^{13}$C-glucose. The complex was formed by titrating unlabeled RNA into $^{13}$C, $^{15}$N labeled protein, and monitored by recording the $^{15}$N HSQC of the protein. NMR experiments were performed on Bruker Avance 600 and Avance 800 spectrometers equipped with HCN cryo-probes and pulsed field gradients. NMR data were processed with NMRPipe[42] and analyzed with CCPNMR[43]. 2D F1-, F2-filtered NOESYs[44] (mixing times 100 and 300 ms) and 2D TOCSY (mixing time 80 ms) were collected on unlabeled RNA in complex with $^{13}$C, $^{15}$N labeled protein in D$_2$O at 298 K to assign the aromatic and sugar protons of the RNA. 2D $^{15}$N/$^{13}$C HSQCs, 3D HNCACB, CBCA(CO)NH, HNCO, HN(CA)CO, HBHA(CO)NH and HCCH-TOCSY spectra were collected on $^{15}$N, $^{13}$C labeled protein in complex with unlabeled RNA to assign the backbone and the non-aromatic side-chains of the protein. 2D $^{13}$C HSQC with carbon centered at -125 ppm and 3D $^{13}$C NOESY-HSQC (mixing time 150 ms) were also recorded to assign aromatic side chains of the protein.

## NMR resonance assignments and structural calculation

Protein assignments followed the regular protocol using triple resonance spectra listed above. RNA resonance assignments were obtained from 2D NOESY and TOCSY by comparison with chemical shifts of GCAUG motif of pre-miR20b stem loop complexed with RbFox and those reported for UGCAUGU-RbFox complex[19,25]. Manually assigned intermolecular NOE distance restraints derived from 3D NOESYs at 100 ms mixing time were separated into three ranges based on the cross-peak intensities, strong (1.8–3.5 Å), medium (1.8–4.5 Å) and weak (1.8–5.5 Å). Additional NOEs observed only in 2D NOESY with 300 ms of mixing time were assigned as very weak (1.8–6.5 Å).

The NOE distance restraints for the complex were obtained from the combination of: intra-protein NOEs automatically assigned from $^{15}$N and $^{13}$C NOESY-HSQCs obtained by CYANA; intra-RNA NOEs

manually assigned from 2D F1-, F2-filtered NOESYs spectra; and inter-molecular NOEs manually assigned from 2D F1-filtered, F2-edited NOESY and 3D $^{13}$C F1-filtered, F3-edited NOESY-HSQC.

Dihedral angle restraints for the conformation of sugar rings (C2'-endo or C3'-endo) were added, based on H1'-H2' cross-peak intensities in 2D TOCSY spectrum. With all of the restraints above, 100 initial structures were generated in CYANA[45], and the 20 structures with the lowest target function were further regularized in an implicit solvent model using the SANDER module of Amber 14.0[46]. We have used ff99bsc0xOL3 force field for RNA and ff14SB force field for protein. The script for the restrained simulated annealing protocol was modified from Tolbert et al.[47]. Protein torsion angles were obtained by TALOS+[48]. For the complex, we heated the system to 1500 K during Amber simulated annealing refinement. The 20 lowest-energy structures were analyzed with PROCHECK-NMR[49].

## Comparison of inherent specificity of multiple RBPs

We utilized RbFox HiTS-Eq data as well as RNA Bind-n-Seq (RBNS) datasets from the ENCODE consortium to generate affinity distributions. First, we normalized HiTS-Eq RbFox global binding affinity data to a scale of 0 to 1. In addition, for RBNS datasets, we sorted through the RBPs analyzed by the ENCODE consortium to select only the RBPs that had a single RNA binding domain in their structures or; if the RBP had multiple RBDs, then a single RBD was purified and examined. Specifically, we selected 5 RBPs (PUM1, DAZ3, RBM6, SRSF8, and SRSF11) to generate the affinity distributions. Next, the 5-mer RBNS enrichment ($R$) scores for the selected RBPs were acquired from ENCODE data portal (https://www.encodeproject.org). The R scores were then normalized to a scale of 1-to-Euler's constant ($e$), followed by natural log-normalization to fit R scores to a scale of 0 to 1. The normalized affinities for 5-mer sequence variants were then used to generate the affinity distributions for the six selected RBPs. While there is no agreed-upon method to quantitatively define the inherent specificity, we used the ratio between the highest affinity value to the median affinity value.

## Reporting summary

Further information on research design is available in the Nature Portfolio Reporting Summary linked to this article.

## Data availability

All data generated or analyzed in this study are included in the main text or the supplementary materials. Atomic coordinates and NMR data of RbFox complexed with 5'-UGCAUAU have been deposited in the Protein Data Bank under entry ID 7VRL and Biological Magnetic Resonance Bank under entry ID 36452. One other published PDB code cited in this paper is 2ERR. The source data are provided as a Source Date file with this paper. Source data are provided with this paper.

## Code availability

Perl scripts used for Illumina sequencing data analysis is available on GitHub (https://github.com/hsuanchunlin/HiTS-EQ, https://github.com/xxy103/HiTS-EQ).

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

## Acknowledgements

We thank Michael E Harris, Hsuan-Chun Lin (Univ. Florida, Gainsville, FL) and Ulf-Peter Guenther (DKMS, Dresden, Germany) for advice and technical assistance. We also thank the staff members of the National Facility for Protein Science in Shanghai, Zhangjiang Laboratory for providing technical support and assistance in NMR. This work was supported by the NIH (GM118088 to E.J., GM126942 to G.V.), the National Natural Science Foundation of China (31900863 to F.Y.), the Natural Science Foundation of Heilongjiang Province of China (LH2021C049 to F.Y.), and the funds from the International Cooperation Division at Harbin Institute of Technology.

## Author contributions

X.Y. and W.Y. performed the experimental studies. S.Y. and Y.Z. were involved in data analysis and interpretation. X.Y., W.Y., G.V., E.J., and F.Y. wrote the paper. G.V., E.J., and F.Y. supervised the work and finalized the paper.

## Competing interests

G.V. is a co-founder of Ithax Pharmaceuticals and Ranar Therapeutics. E.J. is a co-founder of Bainom Inc. E.J. is also a current employee of Moderna Therapeutics. However, neither entity has a specific financial interest in the reported studies. The remaining authors declare no competing interests.
