## [Peer Review File · Nature Communications]

REVIEWER COMMENTS

Reviewer #1 (Remarks to the Author):

The paper from Ye et al investigates the binding affinity of the RRM of the splicing factor RbFox using high-throughput sequence methods as well as structural biology by NMR. The authors find that the RRM of RbFox presents a very unusual binding profile whereas only sequence containing the GCAUG sequence can bind with nanomolar affinity while sequence without this pentamer immediately reduce the affinity drastically (several order of magnitude). This create a huge gap in affinity between cognate and non-cognate RNA binders. They found that this gap is less pronounced when a mutant protein is used, since the affinity for non-cognate RNA increase compared to WT while the affinity of the cognate RNA decreases as expected. Finally, the authors went on the determine the structure of the RRM of RbFOX with an RNA containing a GCAUA sequence and found that a rearrangement is taking place affecting region of the protein outside the RNA binding site which may act as a sensor for non-cognate binding.

The paper presents a number of interesting features like the use the HT sequencing to analyse the binding of the WT and mutant RRM and the structure of the RRM bound to a mutant RNA. However, I am not sure if we really learned new aspects in protein-RNA interaction or any new biology associated with RbFox.

Major point:

1_My main concern is on the fact that the author claims the "extraordinary" feature of RbFox RRM with this main gap in affinity between cognate and non-cognate RNA. This aspect in my view was already seen in the original structural study of RbFox RRM bound to UGCAUGU (Auweter et al, 2006)(ref17) which showed that the RRM binds 7 nucleotides with the first 4 and the last 3 binding independently. When one of the specific binding nucleotide is changed in one or the other binding sites, then the affinity drop from nanomolar to micromolar. This was already shown in table II of this 2006 paper with the kD measurement of G2A, C3U, A4I and G5A. What is extraordinary is that a single RRM can bind seven single-stranded nucleotides with such a strong affinity. I think that the binding of Fox RRM should be presented this way, as two independent binding sites: a canonical one binding UGU and a non-canonical one binding UGCA. Also, in the analysis of the sequences, it would be interesting to know if the G2-A4 base-pair induced by RNA binding can be replaced by an other base-pair or not.

2_The structure of the complex solved with the G to A RNA is interesting but reflects only one aspect of the binding of the RRM of FOX (the canonical one). To be complete, the author should then do a structure with the other part of the binding site mutated (for example G2A or A4I) to see if movement in the same part of the RRM or elsewhere is also seen when this other binding site is mutated. This is essential to really complete the study and support the generality of the claims.

Minor point:

3_The Burge lab published RBNseq work on FOX, it would be interesting to compare the RBNseq and the methods shown here.

Reviewer #2 (Remarks to the Author):

In "Two distinct binding modes provide the RNA binding protein RbFox with extraordinary sequence specificity" Ye and colleagues present an exhaustive investigation into the specificity of RbFox. While the problem has been examined in prior work, the exquisite degree of quantitative precision reported here provides a useful paradigm to the field of

RNA-protein interactions. The combination of kinetics, global analyses conducted with methods pioneered by the Jankowsky group, and biophysical measurements integrate into a study that is vastly more than the sum of its parts. Furthermore, the work is highly significant given the pervasive occurrence of RRM throughout nature. I have minor suggestions intended to strengthen an already excellent manuscript and to broaden the scope of the conclusions by linking them to related findings in a different system.

- Page 7, can the authors comment on why the models fail to describe the specificity of the mutant RbFox as well as wild-type on page 8. This is left somewhat nebulous. Is it simply due to the fact that broadened specificity is inherently challenging to model?

- Can the authors expand on the question of if re-wiring of specificity comes at the expense of sequence discrimination. Is this generally the case? I am reminded of a conceptually similar experiment where similar results were obtained in that for dozens of specificity variants in the PUF domain (A protein-RNA specificity code enables targeted activation of an endogenous human transcript, NSMB 2014). The salient finding was the specificity was broadened for non-cognate sequences relative to wild-type in the vast majority of cases. In my mind, this appears to be an important theme.

- Might the presence of protein-partners have profound influences on the binding of the non-cognate sequences? This possibility is raised in the context of LASR but warrants additional discussion as the RNPs are presumably the biologically relevant form of most RBPs. I am again reminded of two RNPs – LST-1/FBF and CPB-1/FBF where partnerships can both expand and restrict specificity depending on the identity of the binding partner (A crystal structure of a collaborative RNA regulatory complex reveals mechanisms to refine target specificity eLife 2019, Cooperativity in RNA-protein interactions: global analysis of RNA binding specificity, Cell Rep 2012, and, A crystal structure of a collaborative RNA regulatory complex reveals mechanisms to refine target specificity eLife 2019). It seems likely that similar plasticity exists in RRMs and is an important question to ponder moving forward.

- Title page RNA-binding should be hyphenated. This is also missed in the manuscript. I would suggest a search for "RNA binding" for correction as needed.

- Page 20 remove the space between 1.8 A-3.5 A

Reviewer #3 (Remarks to the Author):

In this manuscript, Ye et al set out to determine the mechanisms by which the RBFOX family proteins exhibit remarkable specificity to essentially two sequence motifs GCA U/C G. As the authors point out, this specificity has been observed repeatedly in several in vitro SELEX-like methods and in vivo approaches. The major conclusion of this work is that RBFOX2 employs two structurally and functionally distinct mechanisms, one which is favorable to GCAUG and the other mechanism which can bind all other RNA sequences with significantly lower affinity. This claim encapsulates the novelty of this work, as all other aspects of the findings have been observed by multiple other groups using similar methods.

Major comments:

The authors claim that RBFOX is unique among RBPs in this specificity. The authors, however, do little in the way of demonstrating this uniqueness in specificity even though there are multiple studies that provide suitable datasets (as cited in their paper) to demonstrate this claim.

It stands to reason that the following: "The first exclusively accommodates cognate and

closely related RNAs with high affinity. The second mode accommodates all other RNAs with reduced affinity by imposing large thermodynamic penalties on non-cognate sequences" would be largely applicable to any RNA or DNA binding protein with selectivity. The definition of "cognate and closely related RNAs" is unclear. Selecting single RRM-containing proteins with an equally complex motif (RBFOX2's motif contains all 4nts so an accurate comparator would be another RBP with similarly complex motif, that is, not a mono- or di- nucleotide binder) and demonstrating that these RBPs do not show bi-modality in their kmer affinity landscape would help solidify the claim that RBFOX is unique. The U1A RRM may be a good place to start. If RBFOX is not unique, that too would be of significance as the described mechanism would apply more broadly to other proteins.

A great aspect of the author's study their correlation of HiTS-Eq modeled affinities vs affinities measured for individual RNAs (Fig1F). The authors should add more discussion in the results section about this analysis. As shown in Fig1F, the relative affinities correlate very well for the top binders as would be expected as the data is calibrated using the top kmer. However, lower affinity binders correlated mostly by rank and are quite different among experiments on individual RNAs vs HiTS-Eq. Given the author's claims regarding thermodynamics, these results should be more thoroughly discussed. Finally, a central claim is that RBFOX uses structurally & functionally distinct mechanism to bind top motifs vs all others, however in Fig1F, the authors apply competition assays (against the top kmer) to measure the affinities of suboptimal RNAs. Given the distinct mechanisms of binding for cognate and noncognate proposed, is this the best way to derive these affinities. Why not SPR, as was done in Ref2? Table 2 should be modified to include actual values for affinity measurements as well as the rel. affinities in HiTS-Eq. Additionally, can the authors please show their imaged gels related to "Equilibrium binding with individual RNAs."

The author's analysis into how individual nts within motifs are not entirely independent is an important and nuanced assessment of specificity that is often overlooked in the RNA field. Indeed, a PWM, does not fully describe specificity even when it comes to one of the most selective RBPs. The authors should consider adding findings in Fig.2 to the abstract.

The authors investigate a version RBFOX with 4 point mutations using HiTS-Eq. This mutant was engineered in a previous study by the authors themselves (Ref18) to promote binding to a miRNA. Overall, the data on this mutant are exceptionally interesting but the results as presented are not well-justified, feel somewhat out of place, and are not further discussed in the context of NMR studies. How do these mutations relate to "The first exclusively accommodates cognate and closely related RNAs with high affinity. The second mode accommodates all other RNAs with reduced affinity by imposing large thermodynamic penalties on non-cognate sequences." To strengthen their claims on the wt RBFOX's divergent binding mechanisms the authors could perform similar NMR studies on the mutant as a test case which has lost these divergent binding modes. These results may further support that there is a thermodynamic penalty associated with non-cognate binding to RBFOX2. For example, is it possible to determine if the mutant RBFOX2 apo structure is similar to non-cognate RNA bound-RBFOX2 structure?

Of note, the mutant carries R118D mutation, and R118 is one of the residues highlighted that losing contacts with noncognate vs cognate. Can the authors add discussion on this in the results section? How do the mutations that make RBFOX less selective relate to the chemical shifts observed with noncognate RNA for wt RBFOX.

The selected noncognate is UGCAGAU, can the authors explain where this kmer lies within the affinity landscape of RBFOX2, what's the rank? While the G-A change makes a big impact, if it is still in the top 5% bound kmers, is it representative of all other RNAs that are bound via the second mode?

A major gap in this study is how these findings apply to RNA regulation in cells. Similar studies like those presented by Begg et al., NSMB (2020) highlight the affinity landscape of RBFOX in vitro and in cells and contextualize it in terms of splicing RNA regulation and organismal development. Adding cell-based assessments would strengthen this study and highlight the importance of their findings. How do their affinity measurements relate to regulation of the same motifs in cells?

Given the challenges the authors describe interpreting their NMR data with the suboptimal kmer and having to borrow data from Ref 18 to obtain reliable structural ensembles, the authors should consider follow-up biochemical validation, perhaps by mutagenesis. While I think the claim that there are two functionally and structurally distinct binding modes by RBFOX, the underlying mechanistic NMR study would benefit from validation.

Minor comments:

Figures are clear and concise but adding more information to certain sections of the main results text would help lead the reader through this study. More on the mutant RBFOX needs to be included as it is the context of one complete figure.

Point-by-point response to the reviewers' comments:

*The reviewers' comments are in *Italic*. Author's responses are in green. All revisions are highlighted in red in the revised manuscript.

Reviewer #1 (Remarks to the Author):

The paper from Ye et al investigates the binding affinity of the RRM of the splicing factor RbFox using high-throughput sequence methods as well as structural biology by NMR. The authors find that the RRM of RbFox presents a very unusual binding profile whereas only sequence containing the GCAUG sequence can bind with nanomolar affinity while sequence without this pentamer immediately reduce the affinity drastically (several order of magnitude). This create a huge gap in affinity between cognate and non-cognate RNA binders. They found that this gap is less pronounced when a mutant protein is used, since the affinity for non-cognate RNA increase compared to WT while the affinity of the cognate RNA decreases as expected. Finally, the authors went on the determine the structure of the RRM of RbFOX with an RNA containing a GCAUA sequence and found that a rearrangement is taking place affecting region of the protein outside the RNA binding site which may act as a sensor for non-cognate binding.

The paper presents a number of interesting features like the use the HT sequencing to analyse the binding of the WT and mutant RRM and the structure of the RRM bound to a mutant RNA. However, I am not sure if we really learned new aspects in protein-RNA interaction or any new biology associated with RbFox.

We appreciate the careful reading of our manuscript and the encouraging assessment of our approaches. Regarding the skepticism concerning new insight into RNA-protein interactions and new biology of RbFox: our manuscript shows, for the first time, two functionally and structurally distinct binding modes for an RBP. That is an entirely novel concept and, as we also demonstrate, key to the extraordinary sequence specificity of RbFox. Our manuscript shows that RBPs can alter their structure in response to minute changes in the RNA sequence (even at regions distant from the RNA binding site) and thereby achieve specificity that is physically unattainable by a single binding mode. Considering these foundational insights, we feel that we have in fact learned fundamental new aspects of RBP function.

To further strengthen our response, we cite the comments of reviewer #2: "*While the problem has been examined in prior work, the exquisite degree of quantitative precision reported here provides a useful paradigm to the field of RNA-protein interactions ... Furthermore, the work is highly significant given the pervasive occurrence of RRMs throughout nature*".

Major point:

1_ My main concern is on the fact that the author claims the "extraordinary" feature of RbFox RRM with this main gap in affinity between cognate and non-cognate RNA. This aspect in my view was already seen in the original structural study of RbFox RRM

bound to UGCAUGU (Auweter et al, 2006)(ref17) which showed that the RRM binds 7 nucleotides with the first 4 and the last 3 binding independently. When one of the specific binding nucleotide is changed in one or the other binding sites, then the affinity drop from nanomolar to micromolar. This was already shown in table II of this 2006 paper with the kD measurement of G2A, C3U, A4I and G5A.

The point of our manuscript was not to simply show, but to explain the affinity difference between cognate and non-cognate sequences. The previously reported large difference in affinity between cognate and non-cognate sequences motivated us to investigate the molecular basis for this phenomenon. We prominently noted this rationale in the introduction to our manuscript, we cited the manuscript, and even compared our data with those previous data.

What is extraordinary is that a single RRM can bind seven single-stranded nucleotides with such a strong affinity. I think that the binding of Fox RRM should be presented this way, as two independent binding sites: a canonical one binding UGU and a non-canonical one binding UGCA.

We followed the reviewer's suggestion and carefully examined our analysis of all binding site variants for notable distinctions between the two binding sites proposed by the reviewer. However, we did not find evidence for significant differences between the two proposed sites; mutations in either site have similar effects on RNA binding. Thus, while our data reveal a distinct binding mode for the cognate and a closely related sequence, no functional differences between the two proposed binding sites are seen. For this reason, we prefer to keep the current focus of presentation of RNA binding by the RbFox RRM.

Also, in the analysis of the sequences, it would be interesting to know if the G2-A4 base-pair induced by RNA binding can be replaced by another base-pair or not.

We thank the reviewer for raising this point. The importance of G2-A4 base-pair within 5'-UGCAUGU is revealed in the global distribution plot as shown in **Fig. 1g**, **Fig. 2e** and **Supplementary Fig. 4a**, substituting the G2-A4 interactions lowers the binding affinity dramatically. In the structure of the RbFox-UGCAUGU complex, A4 does not form any direct H-bonds with RbFox, but interacts with G2. Thus, this provides an ideal opportunity to investigate the impact of G2-A4 base-pair on binding affinity. Following the reviewer's suggestion, we analyzed the binding profiles to RbFox for 5'-GCNUG sequences. The relative affinities for 5'-GCNUG variants show the affinities decrease when changing the A nucleotide at the fourth position. Thus, the G2-A4 base-pair stabilized by RNA binding cannot be replaced by another base-pair.

We included the above mentioned findings on page 7 of the manuscript to underline the importance of using the pairwise coupling model, and added a new **Supplementary Fig. 5** to present the analysis of the binding affinities to RbFox for 5'-GCNUG sequences, as below:

Supplementary Figure 5. Analysis of the binding affinities to RbFox for 5'-GCNUG sequences.

Relative affinities ($K_{A,rel}$) for selected 5-mer RNA variants, indicated on the below. 48 $K_{A,rel}$ values correspond to each 5-mer (vertical line: median; box: variability through lower quartile and upper quartile; whiskers: variability outside the lower and upper quartiles).

2_ The structure of the complex solved with the G to A RNA is interesting but reflects only one aspect of the binding of the RRM of FOX (the canonical one). To be complete, the author should then do a structure with the other part of the binding site mutated (for example G2A or A4I) to see if movement in the same part of the RRM or elsewhere is also seen when this other binding site is mutated. This is essential to really complete the study and support the generality of the claims.

As noted in the response to point #1, we did not find evidence for functional differences between the two proposed sites; mutations in either site have similar effects on RNA binding. Thus, we believe that our analysis of the UUCAUGU variant supports the generality of our main point: the existence of two functionally and structurally distinct binding modes of RbFox.

However, following the reviewer's suggestion, we titrated an RNA variant where G2 was mutated to U, and observed chemical exchange on intermediate to fast timescale for the G2->U variant binding to RbFox, suggesting much weaker binding compared to the cognate RNA bound to RbFox; this is consistent with our bimodal hypothesis. The ^1H - ^{15}N HSQC shows highly inhomogeneous peak intensities, consistent with an irregular binding pocket. In addition, many protein HN resonances located around the direct binding region, such as N123, F126 and F129, disappear during the titration, indicating the interaction is too weak (intermediate exchange regime) for NMR structural

determination. Despite these challenges, we were able to assign most of the HN resonances and compare the chemical shift differences between RbFox protein bound to G2->U RNA variant and to the cognate RNA, as we did for the G6->A RNA variant. As shown in the revised **Fig. 5**, we observe a similar pattern of chemical shift changes in the non-cognate binding mode: major changes in the direct binding loops, and minor changes in the distal site ($\alpha 1/\beta 2$ and $\beta 3/\alpha 2$ loop and the $\alpha 1$ on the protein back side). Mapping of these differences onto the structure of RbFox protein revealed four major regions directly contacting G2 and its base pairing partner A4: $\beta 1$ and its following loop (L119-R129), $\beta 3$ and its preceding loop (I149-F160), and $\beta 4$ and its preceding loop (G178-V188). Importantly, the G2->U mutation also affects the distal $\alpha 1/\beta 2$ and $\beta 3/\alpha 2$ loops of RbFox to a similar extent as the G6->A mutation, suggesting that, very likely, the G2->U variant induces a similar conformational rearrangement in RbFox involving the backside alpha helices as G6->A does. The corresponding data are shown in the revised **Fig. 5**, **Supplementary Fig. 10**, and outlined in the text (page 10-11), as also reported below:

*“Next we collected 3D NMR spectra at saturating RNA concentrations and calculated the chemical shift difference (CSD) between the RbFox complexes bound to the consensus RNA and to RNA1 and RNA2 (**Fig. 5a, d**). For RNA1, mapping of these differences onto the structure of wt RbFox identified changes in four regions of the protein: $\beta 1$ (R118-V121), $\beta 2$ and the following loop (I143-E152), $\beta 3$ (G159-E164) and the C-terminal tail (A191-A193) (**Fig. 5c**). In the structure of RbFox RRM complexed with its cognate RNA, the AUG (nucleotides 4 to 6) element is bound in a canonical manner through π - π stacking with the β -sheet surface, while UGC (nucleotides 1 to 3) is recognized by loop residues. Accordingly, similar chemical shifts for the region of RbFox interacting with UGC were observed in the two complexes, indicating that the UGC nucleotides in both structures maintain the same contacts within the RNA-binding cleft of RbFox. In contrast, the G->A substitution at position 6 causes marked changes in the β -sheet region that binds to the AUG element (0.2 - 0.7ppm, **Fig. 5a**). We also observed significant chemical shift changes (> 0.1ppm) in the $\alpha 1/\beta 2$ and $\beta 3/\alpha 2$ loops (highlighted and labeled in blue in **Fig. 5a, b**), which are distant from the RNA-binding surface (**Fig. 5c**). For RNA2, where G2 is mutated to U, we observed a similar pattern of CSDs, not only in the direct RNA-binding cleft, but also on the distal $\alpha 1/\beta 2$ and $\beta 3/\alpha 2$ loops (**Fig. 5d, e, f**). Mapping of these differences onto the structure of wt RbFox protein revealed four major regions directly contacting G2 and its base pairing partner A4: $\beta 1$ and its following loop (L119-R129), $\beta 3$ and its preceding loop (I149-F160), and $\beta 4$ and its preceding loop (G178-V188). Compared to the G6->A mutation, the distal $\alpha 1/\beta 2$ and $\beta 3/\alpha 2$ loops of RbFox are affected by the G2->U mutation to a similar extent, although the overall changes induced by the G2->U mutation are much smaller.*

*To gain detailed insight into the structural differences between the RbFox RRM structures with cognate and non-cognate RNAs, we determined the structure of the RbFox RRM with the RNA1 substrate. The ^1H - ^{15}N HSQC and NOESY spectra changed during data collection, suggesting multiple binding orientations that are most likely due to the weak binding of the non-cognate RNA. These characteristics made unambiguous NOE identification difficult. To overcome this problem, when collecting spectra for the protein, we oversaturated the protein with RNA and changed the sample frequently during data collection to ensure RbFox was consistently in the fully bound conformation. For the RNA, we used some NOEs for the GCA sequence from our previous NMR dataset within the pre-miR20b stem loop in complex with RbFox. Since chemical shifts are very similar for both proteins and RNAs, we obtained a dataset that converged to a reliable structural ensemble (**Supplementary Table 1**). RNA2 binds even more weakly, as demonstrated by highly inhomogeneous peak intensities (**Fig. 5e**) and the loss of multiple HN*

signals in the ^1H - ^{15}N HSQC (highlighted in grey in **Fig. 5d**); because of the poor spectral quality resulting from reduced affinity, the structure of RbFox bound to RNA2 could not be determined.”

For the reviewer’s convenience, we paste the revised **Fig. 5** and **Supplementary Fig. 10** here, as below:

Figure 5. NMR analysis of the interaction of RbFox RRM with its consensus RNA and two RNA variants.

a Chemical shift difference (CSD) between RbFox RRM bound to its cognate RNA 5'-UGCAUGU and RNA1 5'-UGCAUUAU. Regions with significant CSD while distant to the RNA-binding site are highlighted in blue.

b Superposition of ^1H - ^{15}N HSQC spectra of RbFox RRM complexed with the two RNAs (red: 5'-UGCAUGU; black: 5'-UGCAUUAU). Residues with significant CSDs distant from the RNA-binding site are labeled in blue.

c Mapping of the CSD in panel **a** onto the structure of RbFox bound to its cognate RNA (pdb #2ERR) (red: CSD > 0.2 ppm; orange: 0.1 ppm < CSD < 0.2 ppm; yellow: 0.05 ppm < CSD < 0.1 ppm).

d CSD between RbFox RRM bound to its cognate RNA 5'-UGCAUGU and RNA2 5'-UUCAUGU. Residues broaden out due to intermediate chemical exchange are highlighted in grey, and residues with significant CSD while distant to the RNA-binding site are highlighted in blue.

e Superposition of ^1H - ^{15}N HSQC spectra of RbFox RRM complexed with the two RNAs in panel **d** (red: 5'-UGCAUGU; black: 5'-UUCAUGU). Residues with significant CSDs distant from the RNA-binding site are labeled in blue.

f Mapping of the CSD in panel **d** onto the structure of RbFox bound to its cognate RNA (pdb #2ERR) (same color scheme as in panel **c**).

Supplementary Figure 10. ^1H - ^{15}N HSQC titrations of RbFox RRM with three RNAs UGCAUGU, UGCAUUAU and UUCAUGU.

a Superposition of ^1H - ^{15}N HSQC spectra obtained with ^{15}N -RbFox RRM and increasing amount of UGCAUGU RNA. The peaks corresponding to the free and RNA-bound RRMs (RRM:RNA ratios of 1:0, 1:0.1, 1:0.2, 1:0.5 and 1:1) are colored as black, red, orange, green and blue, respectively.

b Superposition of ^1H - ^{15}N HSQC spectra obtained with ^{15}N -RbFox RRM and increasing amount of UGCAUUAU RNA. The color scheme is the same as in **a**.

c Superposition of ^1H - ^{15}N HSQC spectra obtained with ^{15}N -RbFox RRM and increasing amount of UUCAUGU RNA. The peaks corresponding to the free and RNA-bound RRMs (RRM:RNA ratios of 1:0, 1:0.4, 1:1.2, and 1:2) are colored as black, green, blue and red, respectively.

Minor point:

3_ The Burge lab published RBNseq work on FOX, it would be interesting to compare the RBNseq and the methods shown here.

We thank the reviewer for this suggestion. We have performed this analysis and show the results in **Supplementary Fig. 4b** ($R^2 = 0.61$; with a bimodal affinity distribution). We cited the reference accordingly.

Reviewer #2 (Remarks to the Author):

In “Two distinct binding modes provide the RNA binding protein RbFox with extraordinary sequence specificity” Ye and colleagues present an exhaustive investigation into the specificity of RbFox. While the problem has been examined in prior work, the exquisite degree of quantitative precision reported here provides a useful paradigm to the field of RNA-protein interactions. The combination of kinetics, global analyses conducted with methods pioneered by the Jankowsky group, and biophysical measurements integrate into a study that is vastly more than the sum of its parts. Furthermore, the work is highly significant given the pervasive occurrence of RRMs throughout nature. I have minor suggestions intended to strengthen an already excellent manuscript and to broaden the scope of the conclusions by linking them to related findings in a different system.

We thank the reviewer for the very positive and supporting words.

- Page 7, can the authors comment on why the models fail to describe the specificity of the mutant RbFox as well as wild-type on page 8. This is left somewhat nebulous. Is it simply due to the fact that broadened specificity is inherently challenging to model?

We thank the reviewer for raising this important point. We had not provided a clear explanation for our reasoning: that the failure of a single model to describe the affinity distribution indicates there are multiple distinct binding modes. We have clarified the corresponding passage in the manuscript:

“Collectively, our analyses of the RbFox-RNA interaction with quantitative binding models suggest that the bimodal affinity distribution of RbFox cannot be explained by a single binding mode. We therefore conclude that RbFox employs distinct binding modes – one for its cognate 5-mer and one or more other modes for non-cognate sequence variants.”

- Can the authors expand on the question of if re-wiring of specificity comes at the expense of sequence discrimination. Is this generally the case? I am reminded of a conceptually similar experiment where similar results were obtained in that for dozens of

specificity variants in the PUF domain (A protein-RNA specificity code enables targeted activation of an endogenous human transcript, NSMB 2014). The salient finding was the specificity was broadened for non-cognate sequences relative to wild-type in the vast majority of cases. In my mind, this appears to be an important theme.

We thank the reviewer for raising this point as well. The ability of RNA-binding proteins to impose a high thermodynamic penalty on non-cognate sequences might be diminished in a similar manner in other RBPs, such as those noted on the highlighted study. We have now highlighted the potentially broad applicability of this theme in the text and cited the manuscript noted by the reviewer, in the third paragraph of the Discussion section:

“Related observations were reported for the RBP FBF-2, a member of the PUF-protein family, where mutations in the RNA-binding site increased accommodation for multiple sequence variants.”

- Might the presence of protein-partners have profound influences on the binding of the non-cognate sequences? This possibility is raised in the context of LASR but warrants additional discussion as the RNPs are presumably the biologically relevant form of most RBPs. I am again reminded of two RNPs – LST-1/FBF and CPB-1/FBF where partnerships can both expand and restrict specificity depending on the identity of the binding partner (A crystal structure of a collaborative RNA regulatory complex reveals mechanisms to refine target specificity eLife 2019, Cooperativity in RNA-protein interactions: global analysis of RNA binding specificity, Cell Rep 2012, and, A crystal structure of a collaborative RNA regulatory complex reveals mechanisms to refine target specificity eLife 2019). It seems likely that similar plasticity exists in RRMs and is an important question to ponder moving forward.

We agree with the reviewer once again; structural plasticity in RBPs (along with post-translational modifications) is likely a universal theme in RNA biology. For RbFox, specific conclusions for RbFox, however, have to await future studies of RbFox-RNA binding with co-factors. We cited “A crystal structure of a collaborative RNA regulatory complex reveals mechanisms to refine target specificity eLife 2019, Cooperativity in RNA-protein interactions: global analysis of RNA binding specificity, Cell Rep 2012” in Introduction, since they provide good examples of works exploring RNA binding specificity.

- Tittle page RNA-binding should be hyphenated. This is also missed in the manuscript. I would suggest a search for “RNA binding” for correction as needed.

We followed the advice and added hyphenation between RNA and binding throughout the text.

- Page 20 remove the space between 1.8 A-3.5 A

We have changed the text accordingly.

Reviewer #3 (Remarks to the Author):

In this manuscript, Ye et al set out to determine the mechanisms by which the RBFOX family proteins exhibit remarkable specificity to essentially two sequence motifs GCA U/C G. As the authors point out, this specificity has been observed repeatedly in several in vitro SELEX-like methods and in vivo approaches. The major conclusion of this work is that RBFOX2 employs two structurally and functionally distinct mechanisms, one which is favorable to GCAUG and the other mechanism which can bind all other RNA sequences with significantly lower affinity. This claim encapsulates the novelty of this work, as all other aspects of the findings have been observed by multiple other groups using similar methods.

Major comments:

The authors claim that RBFOX is unique among RBPs in this specificity. The authors, however, do little in the way of demonstrating this uniqueness in specificity even though there are multiple studies that provide suitable datasets (as cited in their paper) to demonstrate this claim.

The reviewer raises a good point. We have examined the global affinity distribution for five RBPs from ENCODE RBNS datasets. RbFox has extraordinary specificity compared to other RBPs. In addition, the RbFox global affinity distribution shape is distinct compared to C5 and hnRNPA1 protein our lab and the Blanton lab studied^{1,2}.

The inherent specificity of RNA-binding protein refers to the ability of an RBP to distinguish one RNA binding site from another when only the interactions between RBP and RNA sequence variants are considered. Many in vitro RNA-RBP affinity assays have shown that the inherent specificities of RBPs are diverse. One way to visualize the differences in the inherent specificity is to consider the affinity distributions for the RNA-RBP interactions.

To do so, we utilized RbFox HiTS-Eq data as well as RNA Bind-n-Seq (RBNS) datasets from the ENCODE consortium to generate the affinity distributions. First, we normalized HiTS-Eq RbFox global binding affinity data to a scale of 0-to-1 (**Supplementary Fig. 4a**). In addition, for the RBNS datasets, we sorted through the RBPs analyzed by the ENCODE consortium to select only the RBPs that had only a single RNA binding domain in their structures or, for multi-domain RBPs, had only a single RBD purified. Specifically, we selected five RBPs (PUM1, DAZ3, RBM6, SRSF8, and SRSF11) to generate the affinity distributions. Next, the 5-mer RBNS enrichment (R) scores for the selected RBPs were acquired from the ENCODE data portal. The R scores were then normalized to scale of 1-to-Euler's constant (e), followed by natural log-normalization to fit the R scores to a scale of 0-to-1. The normalized affinities for 5-

mer sequence variants were then used to generate the affinity distributions for the six selected RBPs. The differences in the shapes of the affinity distributions of the six RBPs show how diverse the inherent specificities of different RBPs are.

While there is no agreed-upon method to quantitatively define the inherent specificity, one useful method involves looking at the ratio between the highest affinity value to the median affinity value. Given the skewed nature of many of the affinity distributions, the median affinity value gives a good representation of the whole affinity distributions. The ratio of the maximum to median affinity values for the six RBPs show that RbFox indeed has a very high ratio between the maximum and the median affinities, indicating that the RbFox can strongly distinguish highest affinity sequence motif from the other 5-mer sequence variants.

We included these results in the first Result section on page 6, added a new figure as **Supplementary Fig. 3**, and updated “Comparison of inherent specificity of multiple RBPs” in Methods.

a**b**

RBP	Affinity Ratio (max/median)
RbFox	31.5
PUM1	12.4
DAZ3	7.3
RBM6	4.3
SRSF8	3.0
SRSF11	1.5

Supplementary Figure 3. Affinity distributions and ratios for six RBPs.

a Global affinity distributions for six RBPs (RbFox, PUM1, DAZ3, RBM6, SRSF8, and SRSF11) from HiTS-Eq data and ENCODE RBNS datasets.

b Affinity ratios between the highest affinity value to the median affinity value for the six RBPs.

It stands to reason that the following: “The first exclusively accommodates cognate and closely related RNAs with high affinity. The second mode accommodates all other RNAs with reduced affinity by imposing large thermodynamic penalties on non-cognate sequences” would be largely applicable to any RNA or DNA binding protein with

selectivity. The definition of “cognate and closely related RNAs” is unclear. Selecting single RRM-containing proteins with an equally complex motif (RbFOX2’s motif contains all 4nts so an accurate comparator would be another RBP with similarly complex motif, that is, not a mono- or di- nucleotide binder) and demonstrating that these RBPs do not show bi-modality in their kmer affinity landscape would help solidify the claim that RbFOX is unique. The U1A RRM may be a good place to start. If RbFOX is not unique, that too would be of significance as the described mechanism would apply more broadly to other proteins.

The reviewer makes yet another good point. We developed pairwise coupling models for four other RBPs from the ENCODE RBNS datasets, and observed single binding modes for them, supporting RbFox is indeed unique.

We show this result as **Figure R1** as below:

Figure R1. Analysis of RNA binding of four other RBPs with pairwise coupling binding models.

Correlation between affinity values from RBNS dataset and values predicted with the PWC binding model for four RBPs: PUM1, RBM6, SRSF8, and SRSF11.

A great aspect of the author’s study their correlation of HiTS-Eq modeled affinities vs affinities measured for individual RNAs (Fig1F). The authors should add more discussion in the results section about this analysis. As shown in Fig1F, the relative affinities correlate very well for the top binders as would be expected as the data is calibrated using the top kmer. However, lower affinity binders correlated mostly by rank and are quite different among experiments on individual RNAs vs HiTS-Eq. Given the author’s claims regarding thermodynamics, these results should be more thoroughly

discussed. Finally, a central claim is that RBFOX uses structurally & functionally distinct mechanism to bind top motifs vs all others, however in Fig1F, the authors apply competition assays (against the top kmer) to measure the affinities of suboptimal RNAs. Given the distinct mechanisms of binding for cognate and noncognate proposed, is this the best way to derive these affinities. Why not SPR, as was done in Ref2?

We thank the reviewer for his/her positive assessment by saying “A great aspect of the author’s study their correlation of HiTS-Eq modeled affinities vs affinities measured for individual RNAs (Fig1F).” We apologize for confusing readers by using both competition assays and SPR. We selected gel shift-based competition assays so the measurements reflect the HiTS-Eq equilibrium binding process.

Table 2 should be modified to include actual values for affinity measurements as well as the rel. affinities in HiTS-Eq. Additionally, can the authors please show their imaged gels related to “Equilibrium binding with individual RNAs.”

We thank the reviewer for pointing out table 2. We now show the RNA sequences in the Source Data file for **Fig. 1f** and **Fig. 3d**. In addition, we show imaged gel examples for two RNA substrates in **Fig. 1a** and **Fig. 3b**, Source Data file for **Fig. 1a**, **Fig. 3b**.

The author’s analysis into how individual nts within motifs are not entirely independent is an important and nuanced assessment of specificity that is often overlooked in the RNA field. Indeed, a PWM, does not fully describe specificity even when it comes to one of the most selective RBPs. The authors should consider adding findings in Fig.2 to the abstract.

We thank the reviewer for recognizing another important aspect of our work. Indeed, considering coupled contributions from multiple positions in the binding site helps draw a more complete picture of the specificity of RNA-binding proteins. Taking the G2-A4 base-pairing interaction as an example, the importance of the G2-A4 base-pair within 5'-UGCAUGU is revealed in the global distribution plot as shown in **Fig. 1g**, **Fig. 2e** and **Supplementary Fig. 4a**; substituting the G2-A4 interaction lowers the binding affinity dramatically. In addition, to address reviewer #1’s question concerning whether the G2-A4 base-pair induced by RNA binding can be replaced by another base-pair, by analyzing the binding profiles to RbFox for different 5'-GCNUG sequences, we found the affinities decrease when changing the A nucleotide at the fourth position (newly added **Supplementary Fig. 5**). Thus, the G2-A4 base-pair cannot be replaced by another base-pair, highlighting the importance of couplings involving even non-adjacent nucleotides. This concept is indeed important, but not a completely new finding, because similar observations have been made for DNA-binding³ and RNA-binding proteins reported by us and others^{1,4}. Therefore, to help the readers focus on our main findings concerning the distinct structural and functional binding modes of RbFox, we chose not to comment on this aspect of the work in the abstract, but to simply add new text on page 7, as below:

*“Considering coupled contributions from multiple positions within the binding sites often improves the description of the experimental variance. We analyzed the binding profiles to RbFox for 5'-GCNUG sequences (**Supplementary Fig. 5**), and found that substituting the G2-A4 base-pairing interaction lowers the binding affinity dramatically, highlighting the importance of couplings involving even non-adjacent nucleotides.”*

The authors investigate a version RBFOX with 4 point mutations using HiTS-Eq. This mutant was engineered in a previous study by the authors themselves (Ref18) to promote binding to a miRNA. Overall, the data on this mutant are exceptionally interesting but the results as presented are not well-justified, feel somewhat out of place, and are not further discussed in the context of NMR studies. How do these mutations relate to “The first exclusively accommodates cognate and closely related RNAs with high affinity. The second mode accommodates all other RNAs with reduced affinity by imposing large thermodynamic penalties on non-cognate sequences.” To strengthen their claims on the wt RBFOX’s divergent binding mechanisms the authors could perform similar NMR studies on the mutant as a test case which has lost these divergent binding modes. These results may further support that there is a thermodynamic penalty associated with non-cognate binding to RBFOX2. For example, is it possible to determine if the mutant RBFOX2 apo structure is similar to non-cognate RNA bound-RBFOX2 structure?

The four-point mutant of RbFox, designated as RbFox^{mut} in the manuscript, is indeed an idea control to test our thermodynamic penalty model for wt RbFox. As stated on page 8 in the manuscript, our HiTS-Eq data on RbFox^{mut} shows a global increase in relative affinities for non-cognate RNA sequences and a pronounced decrease in specificity, diminishing the pronounced bimodal binding distribution observed with wt RbFox. In this manner, the divergent binding mode of wt RbFox is lost in RbFox^{mut}.

Following the reviewer’s suggestion, we carried out similar NMR analysis to investigate whether wt RbFox’s divergent binding mode is structurally lost in the mutant. We used two RNAs, a 7-nt cognate RNA 5'-UGAAUGU and a non-cognate RNA 5'-UGAAUCU, to perform titrations into RbFox^{mut} and collected ¹H-¹⁵N HSQC spectra, as was done for wt RbFox (**Supplementary Fig. 12a, b**). In contrast to wt RbFox, we observe slow-exchange binding on the chemical shift time scale with both RNAs, indicating both cognate and non-cognate RNAs bind to RbFox^{mut} with strong affinity, better than μ M.

Taking advantage of the tight binding, we were able to collect high quality 3D NMR spectra to complete backbone HN assignments for both RbFox^{mut} complexes and compared the CSDs between them. As shown in **Supplementary Fig. 12c, d, e**, compared to wt RbFox (**Fig. 5**), fewer residues in RbFox^{mut} protein are affected by the single-nucleotide mutation and smaller overall CSDs are observed, suggesting the chemical shift changes in RbFox^{mut} originate mainly from chemical environment differences caused by the nucleotide change, but not by a structural change. Intriguingly, CSDs occurring away from the RNA-binding surface induced by G6->A or G2->U mutations, as observed with wt RbFox, were not observed for the G6->C mutation with RbFox^{mut}. Taken together, these results provide strong evidence that the

conformational divergence induced by RNA substrates observed with wt RbFox is lost in RbFox^{mut}, strengthening our notion that a bimodal binding mode is provided uniquely by wt RbFox.

We revised the manuscript accordingly on page 13; **Supplementary Fig. 12** was also added. For the reviewer's convenience, we paste the newly added **Supplementary Fig. 12** here:

Supplementary Figure 12. Characterization of RbFox^{mut} bound to two RNAs: UGAAUGU and UGAAUCU.

a Superposition of ¹H -¹⁵N HSQC spectra obtained with ¹⁵N-RbFox^{mut} and increasing amount of UGAAUGU RNA. The peaks corresponding to the free and RNA-bound RbFox^{mut}s (RRM:RNA ratios of 1:0, 1:0.2, 1:0.5 and 1:1) are colored as black, red, green and blue, respectively.

b Superposition of ¹H -¹⁵N HSQC spectra obtained with ¹⁵N- RbFox^{mut} and increasing amounts of UGAAUCU RNA. The color scheme is the same as in **a**.

c Chemical shift difference (CSD) between RbFox^{mut} bound to UGAAUGU and UGAAUCU.

d Superposition of ¹H-¹⁵N HSQC spectra of RbFox^{mut} complexed with the two RNAs (black: UGAAUGU; red: UGAAUCU).

e Mapping of the CSDs in panel **c** onto the structure of RbFox-RNA complex (pdb #2ERR) (red: CSD > 0.2 ppm; orange: 0.1 ppm < CSD < 0.2 ppm; yellow: 0.05 ppm < CSD < 0.1 ppm).

Of note, the mutant carries R118D mutation, and R118 is one of the residues highlighted that losing contacts with noncognate vs cognate. Can the authors add discussion on this in the results section? How do the mutations that make RBFOX less selective relate to the chemical shifts observed with noncognate RNA for wt RBFOX.

The reviewer makes yet another good point. We understand the reviewer invites us to compare the chemical shifts between R118 in wt RbFox's complex with its noncognate RNA and D118 in RbFox^{mut}, which might provide some clues as to why RbFox^{mut} is less selective in the RNA substrates than wt RbFox.

Following the reviewer's suggestion, we did not find any correlation between the chemical shifts of D118 in RbFox^{mut} and R118 in wt RbFox's complex with its noncognate RNA. However, we notice that R118 in wt RbFox is special, with H and N resonances both shifted downfield to rare values (9.0 ppm, 129.3 ppm, respectively), compared to those of D118 in RbFox^{mut}, which fall within the normal range (8.2 ppm, 120.8 ppm). This implies a unique structural property in wt RbFox that might favor RNA selectivity.

The selected noncognate is UGCAGAU, can the authors explain where this kmer lies within the affinity landscape of RBFOX2, what's the rank? While the G-A change makes a big impact, if it is still in the top 5% bound kmers, is it representative of all other RNAs that are bound via the second mode?

We thank the reviewer for noticing this point. The UGCAUAU ranks in the 17.4 percentile among 16,384 sequence variants. And the UUCAUGU ranks in the 37.6 percentile among 16,384 sequence variants. Overall, both non-cognate sequence variants we selected represent examples for non-cognate sequence binding modes. We added this information in the Result section of "cognate and non-cognate RNAs induce distinct structures in the RbFox RRM".

A major gap in this study is how these findings apply to RNA regulation in cells. Similar studies like those presented by Begg et al., NSMB (2020) highlight the affinity landscape of RBFOX in vitro and in cells and contextualize it in terms of splicing RNA regulation and organismal development. Adding cell-based assemsents would strengthen this study and highlight the importance of their findings. How do their affinity measurements relate to regulation of the same motifs in cells?

Cell-based investigation would be instructive. Yet, as the reviewer suggests, experiments addressing these questions are not trivial to implement and perform. While we are not in the position to conduct these experiments, we believe that our manuscript will inspire such cell-based inquiries by other groups.

Given the challenges the authors describe interpreting their NMR data with the suboptimal kmer and having to borrow data from Ref 18 to obtain reliable structural ensembles, the authors should consider follow-up biochemical validation, perhaps by mutagenesis. While I think the claim that there are two functionally and structurally distinct binding modes by RBFOX, the underlying mechanistic NMR study would benefit from validation.

We apologize for the confusion caused by our misuse of the word “borrow” and unclear description of how we managed to collect the NMR data. We revised the text to provide a clearer description on page 11 as below:

“To overcome this problem, when collecting spectra for the protein, we oversaturated the protein with RNA and changed the sample frequently during spectra collection to ensure RbFox was consistently in the full bound conformation. For the RNA, we used some NOEs from our previous NMR dataset for the GCA within the pre-miR-20b stem loop in complex with RbFox.”

In brief, the structure of RbFox protein is the key result supporting our conclusion that there are marked structural differences between the two binding modes, including large conformational rearrangements distant from the RNA-binding site. We are very confident with the accuracy of this structure, and validation using biochemical method is unnecessary. The protein-RNA structural calculation was carried out in three steps: the protein alone (in its holo form), the RNA alone (in its holo form); then the two components were annealed together using intermolecular restraints. With 1795 intramolecular NOEs for a 10 kDa protein, the structure of RbFox is rigid enough to maintain its backbone conformation from the first step to the third step, resulting in very limited variation occurring only on the side chains at the direct RNA-binding surface (RMSD < 0.1 Å). The fact that the RNA conformation is less well defined, does not affect our conclusion, and reflects the very real existence of multiple binding orientations due to weak binding. If we somehow defined the RNA conformation better by adding non-experimental observations, we would provide incorrect information to the reader.

Minor comments:

Figures are clear and concise but adding more information to certain sections of the main results text would help lead the reader through this study. More on the mutant RBFOX needs to be included as it is the context of one complete figure.

We followed the reviewer’s advice and added more descriptions (concerning main figures 2, 3, 4 and 5) in the revised manuscript, including text for the mutant RbFox as well.

References

1. Guenther, U. P. *et al.* Hidden specificity in an apparently nonspecific RNA-binding protein. *Nature* **502**, 385–388 (2013).
2. Jain, N., Lin, H., Morgan, C. E., Harris, M. E. & Tolber, B. S. Rules of RNA specificity of hnRNP A1 revealed by global and quantitative analysis of its affinity distribution. *Proc Natl Acad Sci U S A* **114**, 2206–2211 (2017).
3. Stormo, G. D. & Zhao, Y. Determining the specificity of protein-DNA interactions. *Nat. Rev. Genet.* **11**, 751–760 (2010).
4. Jankowsky, E. & Harris, M. E. Specificity and nonspecificity in RNA-protein interactions. *Nat Rev Mol Cell Biol* **16**, 533–544 (2015).

REVIEWERS' COMMENTS

Reviewer #1 (Remarks to the Author):

The authors have done more experiments and did address most of my comments satisfactorily. I am therefore now happy with this revised version of the paper.

Reviewer #3 (Remarks to the Author):

The authors have addressed the concerns presented in my initial review with experiments, analysis, and additional discussion. In my view this manuscript needs no further revision and should be accepted. I thank the authors for their thoughtful responses .